# Reorganization of Atlantic Waters at sub-polar latitudes linked to deep water overflow in both glacial and interglacial climate states.

Dakota E. Holmes[1,2], Tali L. Babila[3], Ulysses Ninnemann[4], Gordon Bromley[1,2], Shane Tyrrell[6,7], Greig A. Paterson[5], Michelle J. Curran[1,2], Audrey Morley[1,2,7*]

[1]Department of Geography, School of Geography, Archaeology and Irish Studies, National University of Ireland Galway, Galway, Ireland.
[2]Ryan Institute for Environmental, Marine, and Energy Research, Galway, Ireland.
[3]Ocean and Earth Science, University of Southampton, National Oceanography Centre, Southampton, United Kingdom.
[4]Department of Earth Science and Bjerknes Centre for Climate Research, University of Bergen, Bergen, Norway.
[5]Department of Earth, Ocean, and Ecological Sciences, University of Liverpool, Liverpool, United Kingdom.
[6]Earth and Ocean Sciences, School of Natural Sciences, National University of Ireland Galway, Galway, Ireland.
[7]iCRAG Irish Centre for Research in Applied Geosciences, Ireland.

*Correspondence to Audrey Morley (audrey.morley@nuigalway.ie)

**Abstract.** While a large cryosphere may be a necessary boundary condition for millennial-scale events to persist, a growing body of evidence from previous interglacial periods suggests that high-magnitude climate events are possible during low cryosphere climate states. However, the full spectrum of variability, and the antecedent conditions under which such variability can occur, have not been fully described. As a result, the mechanisms generating high-magnitude climate variability during low cryosphere boundary conditions remain unclear. In this study, high-resolution climate records from DSDP site 610 are used to portray North Atlantic climate's progression through low ice, boundary conditions of Marine Isotope Stage (MIS) 11c into the glacial inception. We show that this period is marked by two climate events displaying rapid shifts in both deep overflow and surface climate. The reorganization between Polar and Atlantic Waters at subpolar latitudes appears to accompany changes in the flow of deep water emanating from the Nordic Seas, regardless of magnitude or boundary conditions. Further, during both intermediate and low ice boundary conditions, we find that a reduction in deep water precedes surface hydrographic change. The existence of surface and deep ocean events, with similar magnitudes, abruptness, and surface-deep phasing, advances our mechanistic understanding of and elucidates antecedent conditions that can lead to high-magnitude climate instability.

## 1.0 Introduction

A growing body of evidence investigating older interglacials (McManus et al., 1999; Oppo et al., 1998; Past Interglacials Working Group of PAGES, 2016) show that increased surface climate variability in the North Atlantic is typical of many late Pleistocene interglacials (Ferretti et al., 2015; Irvalı et al., 2020; Irvalı et al., 2016; Kandiano et al., 2017; Mokeddem et al., 2014; Sánchez-Goñi et al., 2016) and may

involve deep water circulation changes. For example, evidence from the Bermuda Rise shows that Marine Isotope Stage (MIS) 5e ends with abrupt changes in surface and deep water circulation attributed to changes in deep water organization (Adkins et al., 1997). More recent evidence suggests that episodes of reduced North Atlantic Deep Water (NADW) may be common features of most interglacial climates of the past 450-thousand-years (Galaasen et al., 2020) and that large changes in the distribution of subpolar North Atlantic Waters act as precursor events to glacial inceptions (Alonso-Garcia et al., 2011; Barker et al., 2015; Irvalı et al., 2016; Mokeddem et al., 2014). These observations suggest that high-magnitude climate variability is possible during both glacial and interglacial boundary conditions and may even play a role during the transition between climate states. Thus, it is crucial to establish the conditions under which such variability can occur, and what components of the climate system were involved (antecedents).

This type of high-magnitude climate variability stands in contrast to the low magnitude multidecadal to millennial variability in atmosphere-ocean circulation (Bianchi and McCave, 1999; Bond et al., 2001; Bond et al., 1997; Thornalley et al., 2013b; Thornalley et al., 2009) observed during the Holocene (McManus et al., 1999; Thornalley et al., 2013a). For example, Thornalley et al. 2009 shows that the maximum range in SST variability recorded south of Iceland does not exceed 3°C during the Holocene. Slightly larger shifts in SSTs of up to 4°C were recorded on the Bjorn Drift in the Late Holocene (Came et al. 2007) possibly associated with the slow advance and passage of the subarctic front. These reconstructions are juxtaposed to SST shifts of 6-8°C recorded during glacial periods (e.g., D-O events, Dokken et al. 2013). Postulated mechanisms for generating low magnitude interglacial climate variability include the Atlantic Meridional Overturning Circulation (AMOC), solar variability, and/or volcanism, but their origins remain uncertain (Grossmann and Klotzbach, 2009; Ottera et al., 2010; Sicre et al., 2011). However, there is evidence for the existence of ocean-atmosphere linkages that communicate and amplify relatively small changes in radiative forcing (e.g., total solar irradiance) into a climate signal extending beyond the north-eastern Atlantic region (Ammann et al., 2007; Knudsen et al., 2009; Lockwood et al., 2010; Lohmann et al., 2004; Swingedouw et al., 2010) with potentially global implications (Lean, 2010; Mayewski et al., 2004; Shindell et al., 2001).

Recent evidence suggests that deep water circulation changes preceded abrupt high-magnitude North Atlantic climate events during the last deglaciation (Henry et al., 2016; Muschitiello et al., 2019), Dansgaard-Oeschger (D-O) climate events (Dokken et al., 2013), and may even trigger Heinrich events (H-events) (Alvarez-Solas et al., 2010; Barker et al., 2015; Bassis et al., 2017). However, few records have examined the shorter term (multi-centennial) variability during low cryosphere climate states with a methodology able to discriminate the relative timing of surface climate and deep circulation changes. Our

aim in this study is therefore to investigate the development of North Atlantic climate to changes in Nordic Seas Deep Water (NDW) formation following the low-ice, interglacial boundary conditions of MIS 11. Specifically, we test if NADW circulation changes preceded surface ocean responses, regardless of climate boundary conditions. Considering concerns over industrial era trends in ocean circulation and surface temperature (Caesar et al., 2021; Perner et al., 2019; Rahmstorf et al., 2015), we aim to improve understanding of the stability of the Thermohaline Circulation-climate system and the boundary conditions under which rapid and large perturbations occur.

Preceding the glacial inception, MIS 11 was a particularly long interglacial period and while the northern high-latitude insolation maximum was weaker overall than that of our current interglacial period (MIS 1), due to precession and obliquity being out of phase, the warmest interval of this interglacial (e.g., MIS 11c, defined as in Railsback et al. (2015)) had very similar eccentricity/precession parameters to MIS 1 (Yin and Berger, 2015). Greenhouse gas concentrations, specifically carbon dioxide ($CO_2$), were also similar to pre-industrial values (e.g., 285 ppm at 407.5 kilo annum before present (ka) vs. 280 ppm at 1850 CE) (Etheridge et al., 1996; Macfarling Meure et al., 2006; Nehrbass-Ahles et al., 2020), leading to one of the warmest and longest interglacial periods in the past 800 ka (Irvalı et al., 2020). Summer temperature anomalies at the height of MIS 11 (411ka) were 2.8 ±0.7°C relative to the present. By 403 ka, anomalies were at 0 ±1°C and by 397ka summer temperature anomalies had dropped to -2°C ±1°C relative to the present (Robinson et al., 2017). The similarities between MIS 11c and our current interglacial period make it an interesting analogue for assessing climate variability where boundary conditions only deviate modestly from those experienced today (Mcmanus et al., 2003).

The unusual strength/warmth and long duration of MIS 11c when considering the weak insolation parameters has led to a series of publications discussing potential mechanisms that could have positively amplified the warmth of MIS 11c (Droxler et al., 2003 and chapters therein) A negative feedback involving a stronger Nordic heat pump supported by stronger cross-equatorial heat exchange from the South into the North Atlantic has been proposed as one of the main mechanism explaining this apparent paradox (Berger and Wefer, 2003). The long duration of MIS 11c may also have driven the widespread shrinkage of the Greenland ice sheet (GIS) by 403 ka, with remnants potentially restricted to the east Greenland margin (de Vernal and Hillaire-Marcel, 2008; Robinson et al., 2017; Schaefer et al., 2016). The melting of the GIS also led to unique surface hydrographic conditions in the Nordic Seas, which resulted in lower-than-present SST throughout the first half of the interglacial (Bauch et al., 2000; Doherty and Thibodeau, 2018; Helmke and Bauch, 2003; Kandiano et al., 2012; Kandiano et al., 2016a; Thibodeau et al., 2017). Given the evidence for vigorous deep water formation during the entire interglacial (Dickson et al., 2009; Mcmanus

et al., 2003; Riveiros et al., 2013) Kandiano et al. (2016b) hypothesised that warm Atlantic Waters still reached the Nordic Seas at the subsurface maintaining deep convection perhaps at a more southern and or eastern location when compared to today. Cold and fresh sea surface conditions persisted until ca 410 ka after which warmer sea surface temperatures established until about 400 ka (Kandiano et al., 2016b). At 400 ka, the transition into the boreal insolation minimum set the stage for the glacial inception (Müller and Pross, 2007). At inception (~397 ka), summer insolation decreased to 466 $Wm^{-2}$, but atmospheric concentrations of $CO_2$ remained high and typical of interglacial values (259-265 ppm) (Tzedakis et al., 2012) resulting in one of the warmest inceptions of the past 800 ka most similar to MIS 1 in terms of eccentricity and $CO_2$ levels at the time (Ganopolski et al., 2016; Yin and Berger, 2015).

Similar to the Mid-to-Late Holocene transition, the progression into an insolation minimum at the end of MIS 11 led to a progressive cooling from high (at 400 ka) (Melles et al., 2012; Vogel et al., 2013) to mid-latitudes (at 397 ka) (Kandiano and Bauch, 2007; Rodrigues et al., 2011; Stein et al., 2009; this study). Mechanistically, this would have led to the presence of more sea ice and allowed greater freshwater export into the SPG via the Denmark Strait. This is supported by the onset of sea surface cooling downstream of the EGC, South of Greenland shortly after 398 ka (Irvalı et al., 2020). An increased advection of Arctic Sea ice and subsequent increase in freshwater transport into the subpolar North Atlantic likely freshened surface waters, thereby weakening the cross-gyre density gradient. Over time, these changes in hydrology would have altered ocean frontal positions within the Subpolar Gyre (SPG) (Irvalı et al., 2016), causing them to move eastward and diminish SPG circulation further (Born et al., 2010; Born et al., 2011; Mokeddem et al., 2014). Modern observations and palaeoceanographic investigations also suggest that these hydrographic changes were concurrent with an enhanced inflow of warm Atlantic Waters via the eastern SPG (Born and Levermann, 2010; Morley et al., 2014; Moros et al., 2006; Perner et al., 2015; Risebrobakken et al., 2003) prolonging interglacial warmth at mid-latitudes and providing the moisture necessary for high-latitude ice sheet build-up during the glacial inception (Born et al., 2010; Born et al., 2011). In addition, it is hypothesized that this increased transport of Atlantic Waters could have sustained deep water formation into the insolation minimum (Born et al., 2010; Born et al., 2011; Mokeddem et al., 2014; Thornalley et al., 2009). Thus, our current understanding is that glacial inceptions are marked by distinct changes in surface circulation across the Greenland-Scotland ridge which could alternatively suppress (e.g., via sea ice/freshwater export into the SPG) or sustain (e.g., via Atlantic Water inflow into the Nordic Seas) deep water circulation. However, the sensitivity and impacts of deep circulation changes under such evolving conditions are still poorly understood.

**2.0 Materials and Methods**

## 2.1 Core location and oceanographic setting

Deep Sea Drilling Project (DSDP) site 94-610 (53°13.30' N, 18°53.21' W; 2417 m water depth) is located on the Feni Drift which is a contourite deposit formed by southward flowing of deep water (Ellett and Roberts, 1973; Jones et al., 1970). (Fig. 1). Sediments of the Feni drift are deposited by waning, intermittent bottom currents flowing from North to South along the Feni Ridge because density-driven currents keep bathymetry on their right in the Northern Hemisphere (Johnson et al., 2017). Between episodes of current activity normal pelagic and ice-rafted sedimentation continues unhindered (Robinson and McCave, 1994). At the depth of 2417m, 610B lies within the influence of lower WTOW. The flow pathway for WTOW is around the northern and western boundary of the Rockall Trough (Johnson et al., 2017). Observations demonstrate that the southward flow of deep WTOW is intermittent on annual timescales but positive on $\geq$ decadal timescales (Johnson et al., 2017), which is the resolution that we are targeting in this study. To the north and west of 610B the central anticyclonic gyre of the Rockall Trough (Johnson, 2012; New and Smythe-Wright, 2001; Smilenova et al., 2020), recirculates water down to 2000m during winter mixing (Smilenova et al., 2020). Given the distance from the gyre (ca 500 km) and the deeper depth of 610B, it is unlikely that this influences the sedimentation and flow over the site.

Modern hydrographic observations place DSDP site 610 in the pathway of north-flowing Atlantic Waters at the surface and deep Wyville Thomson Ridge Overflow Waters (WTOW) at depth (Ellet et al., 1986; Johnson et al., 2017). At the surface, the Rockall Trough directs ca. 50% of saline Atlantic Waters into the Nordic Seas (Hansen and Østerhus, 2000) and thus is a main contributor for NDW formation. At depth, the Rockall Trough receives 10%–15% of the NDW spilling over the Wyville Thomson Ridge (Dickson and Brown, 1994; Hansen and Østerhus, 2000). The paired surface and deep water palaeoceanographic reconstructions, therefore, afford a singular vantage to test the relative timing in the response between the surface and deep branch components of the AMOC in the eastern North Atlantic during transitional boundary conditions. We acknowledge that the WTOW carries only a relatively small portion of the total overflows from the Nordic Seas which contribute to the lower branch of AMOC, making it difficult to draw strong conclusions about the integrated overflows from this one location without complementary constraints on deep circulation changes.

## 2.2 Sample preparation

DSDP Holes 94-610A and B were recovered side-by-side near the crest of the Feni Drift during Deep Sea Drilling Project Leg 94 of the *R/V Glomar Challenger*. Continuous high-resolution sediment sampling (every 0.5 cm) focused on Hole B. Here, we present data collected from 212 samples recovered from Core 4, Section 2 between 100 and 150 cm and Section 3 between 0 and 150 cm, corresponding to 26.5–28.495

m below sea floor (mbsf). Each sample had ~1.5 g of sediment preserved for grain size analysis. The remaining sample was placed in distilled water and shaken for 24 hours to disperse the sediment. Once dispersed, samples were wet-sieved and separated into size fractions of > 63 µm and < 63 µm. The > 63 µm fraction was used for the selection of foraminiferal specimens for stable isotope analysis, foraminiferal census counts, and IRD counts after additional dry sieving.

### 2.3 Stable isotopes

We analyzed stable isotopes of the benthic foraminifera species of the genus *Uvigerina*. Tests were picked ca. every 8 cm for a total of 82 samples between 25.73–32.20 mbsf from size fractions > 150 µm (1–2 specimens per analysis). Stable isotope analyses were measured using a Finnigan MAT253 mass spectrometer at FARLAB at the Department of Earth Science and the Bjerknes Centre for Climate Research, University of Bergen. Results are expressed as the average of the replicates and reported relative to Vienna Pee Dee Belemnite (VPDB), calibrated using NBS-19 and crosschecked with NBS-18. Long-term reproducibility (1σ SD) of in-house standards pooled over periods of weeks to months for samples between 10 and 100 mg is better than 0.08‰ and 0.03‰ for $\delta^{18}O$ and $\delta^{13}C$, respectively.

### 2.4 Lithic counts

Relative abundance of Ice Rafted Debris (IRD), an established proxy of ice sheet variability (Baumann et al., 1995; Fronval and Jansen, 1997; Jansen et al., 2000), content in the > 150 µm fraction was counted every 0.5 cm between 26.5–26.915 and 27.12–27.3 mbsf, every 1.0 cm between 26.92–27.11 and 27.31–27.795 mbsf, and every 5 cm between 27.8–28.495 mbsf, for a total of 212 samples. Every 10 samples, IRD in the > 150 µm fraction was counted twice to determine the standard deviation. We compared the IRD counts using a t-test (paired two sample t-test, p < 0.05) and found that there was not a statistically significant difference between the original counts for IRD (M=263.32, SD=390.95) and the recounted IRD samples (M=261.68, SD=389.96) conditions; ($t$(19)=0.9511, $p$=.35). Results are presented as the number of lithogenic/terrigenous grains per gram (grains.g$^{-1}$) of dry sediment.

### 2.5 Foraminiferal counts and species abundance (%)

Planktic foraminiferal assemblages were counted every 0.5 cm for depths 26.5–26.915 mbsf, every 2.5 cm for depths 26.94–27.795 mbsf, and every 5 cm for depths 27.8–28.45 mbsf, for a total of 129 samples. All samples were dry sieved at 150 µm and then split to give at least 300 planktic foraminifera for census counts. The absolute number of planktonic foraminifera counted ranged from 300 to 558. The *N. pachyderma* coiling ratio is calculated as the percentage of *N. pachyderma* of the sum (*N. pachyderma* + *N. incompta*). The coiling ratio is used as a relative SST proxy at high latitudes (Irvalı et al., 2016), since

*N. pachyderma* is abundant in cold/polar regions, whereas *N. incompta* prefers more temperate/subpolar regions.

## 2.6 Sea surface temperature (SST) reconstructions

Here, we employed the modern core-top faunal and SST data from 1,259 samples in the ForCenS database
of curated planktonic foraminifera census counts (Siccha and Kucera, 2017) to reconstruct SSTs over the 26.5—28.495 mbsf interval. The ForCenS database is taxonomically standardised and includes four previously compiled datasets: CLIMAP (Members, 2009), Brown University Foraminiferal Database (Prell et al., 1999), ATL947 (Pflaumann et al., 2003), and MARGO (Barrows and Juggins, 2005; Hayes et al., 2005; Kucera et al., 2005a; Kucera et al., 2005b), as well as six individual datasets (Siccha and Kucera,
2017 and references therein). The higher number of core-tops, and duplicate avoidance in the ForCenS database, increases the quality of the available analogues. We used the ROIJA package (Juggins, 2017) in R (R Core Team, 2019), and a squared chord distance as the dissimilarity measure, to estimate Modern Analogue Technique (MAT)-derived annual SSTs (Hutson, 1980; Prell, 1985) using the WOA98 annual SST dataset. Core tops with dissimilarity greater than 0.4 were not considered. The average dissimilarity
coefficient is 0.108 with a standard deviation of 0.02. The averages of summer and winter SST standard deviations are 2.10°C and 1.65°C, respectively.

## 2.7 X-ray fluorescence

During a visit at the IODP Bremen Core Repository (Germany 03/2017) DSDP Core 94-610A (0–38 m)
and Core 94-610B (24–34 m) were scanned at 0.5 cm resolution using an Avaatech Core Scanner that provides counts for selected elements between Aluminium (Al) and Uranium (U). Here we use the log of calcium over titanium (log (Ca/Ti)) to identify relative changes in biogenic vs. lithogenic contribution at site DSDP 610 (Rothwell, 2015; Solignac et al., 2011). This allowed us to identify interglacial MIS 11 first in DSDP 610A (the archive and best-preserved Hole of the site for the Quaternary period) and subsequently
in DSDP 610B (working Hole of the site). High log (Ca/Ti) values are interpreted to reflect time periods characterized by high productivity and interglacial periods, while low ratios indicate a higher detrital load associated with lower productivity and ice-rafted debris (Lebreiro et al., 2009) typical of stadials (Arz et al., 2001; Gebhardt et al., 2008; Rothwell, 2015).

## 2.8 Grain size analysis
We measured grain size distributions every 0.5 cm and 5.0 cm between 26.5–27.795 and 27.8–28.495 mbsf, respectively when samples had enough sediment available for analysis using a Mastersizer 3000 at the National University of Ireland Galway School of Geography, Archaeology, and Irish Studies. We

applied a refractive index of 1.54 and an absorption index of 0.01, as recommended by *Malvern*
*Panalytical.* Sample batches were prepared following the methodology laid out by (Jonkers et al., 2015)
using approximately 1.5 g of wet bulk sediment, samples were pre-sieved at 1000 μm. Any grains > 1000
μm were measured under a microscope and manually added to the Mastersizer results. The complete grain
size distributions were decomposed statistically using AnalySize (v. 1.2.1) (Paterson and Heslop, 2015)
using non-parametric End Member Analysis (EMA). Both the $R^2$ and angular deviation (θ) goodness-of-
fit statistics indicate that the sediments are adequately described as mixtures of three endmembers ($R^2$ =
0.997, θ = 2.6°; Appendix Fig. A1). The end members are shown in Fig. 2a.

The well-sorted end members 2 (EM2) and 3 (EM3) consist primarily of clay and fine silt sediment with
median grain sizes of 4.68 μm and 8.81 μm, respectively, characteristic of sediments sorted by bottom
water currents. The log ratio of EM2 and EM3 is taken as the proxy of near-bottom current flow strength
(Prins et al., 2002). End member 1 (EM1) is poorly sorted and is composed of 45.91% clay, 39.19% silt,
14.74% sand, and <1% gravel. The presence of sand-sized material in EM1 (> 10 vol %) argues against a
current-sorted origin. Further, > 10% of EM1 are grain sizes ≥ 100 μm, characteristic of IRD supplied by
sea ice. The maximum contributions of EM1 coincide with peak occurrences of IRD supplied by icebergs,
supporting the IRD scenario.

## 2.9 Assessment of uncertainties associated with palaeocurrent proxy log(EM2/EM3):
### 2.9.1 Noise model
Three replicate measurements of 7 specimens and two replicates of a further single specimen were used to
quantity reproducibility of the grain size frequency measurements as measured on a Mastersizer 3000. To
avoid the influence of bubbles present in some of the replicate measurements, the frequency of grain size
bins ≥ 240 μm was set to zero and the measurement was renormalized to sum-to-one. Reproducibility of
the frequency of each grain size bin was quantified as the standard deviation of each bin expressed as a
percentage of the mean frequency of that bin. Where two or more replicates of a grain size bin record zero
frequency, or where no estimate is possible (i.e., for bins with all zeros), the maximum percentage error
across all bins is used, which conservatively overestimates the variance of these bins. The final noise model
is the average percentage standard deviation for the 8 used specimens (Fig. 2b).

### 2.9.2 Confidence intervals
To construct confidence intervals on the end member abundances and their log-ratios we used a Monte
Carlo approach to add simulated noise to the observed grain size frequencies. For the grain size distribution
observation for each specimen, each size bin is randomly resampled from a normal distribution with a

mean given by the observation and a standard deviation prescribed from the noise model described above. Following the addition of noise, each distribution is renormalized to satisfy the non-negative and sum-to-one constraints. We then estimate the abundances of the derived end members using the fully constrained least squares approach (Heinz, 2001). This process is repeated 10,000 times and the 2.5th and 97.5th percentiles are used to define the 95% confidence interval (Fig. 2c). For 10 of our 150 specimens, due to zero abundances of EM2 and/or EM3 in the Monte Carlo analysis, the 95% confidence interval cannot be defined (the presence of zeros leads to undefined values and infinitely bounded confidence intervals). These are highlighted in red in Fig. 2c and correspond to the seven lowest abundances of EM2 and the three lowest abundances of EM3 from the main analysis.

## 2.10 Statistical analyses for quantifying climate transitions

To objectively infer the timing of mean changes observed in the XRF, SST, IRD, and Wyville Thomson Overflow Waters (WTOW) records from DSDP 610B within the studied interval, we use the ramp-fitting method computed in the Fortran 77 program, RAMPFIT (Mudelsee, 2000), which follows the simple approach that the shifts observed in the mean of time series can be characterized as a ramp, i.e., a linear change from one stable state to the other (Appendix Fig. A2). This method estimates the unknown onset and end of a time interval by weighted least-squares regression by a brute-force search. We use a bootstrap simulation of 10,000 resamples to estimate the uncertainty of the results. We took the mean square root of the errors calculated using bootstrap simulation to calculate uncertainties for periods where shifts were not observed. Table 1 shows the climate time-series datasets.

## 3.0 Chronology

The results of the XRF analysis on DSDP Hole 610A (0–38 mbsf) were used to construct a preliminary age model for the past 500 ka and to locate MIS 11 within DSDP 610A and 610B. Scanning of both the archive (DSDP 610A) and the working Hole (DSDP 610B) was necessary for this task due to hiatuses in DSDP 610B preceding MIS11. The age model for DSDP 610A was constrained by assigning a total of 28 calibration points between the log (Ti/Ca) series of DSDP 610A and the LR04 benthic stack record using AnalySeries (v. 2.0) (Paillard et al., 1996). The resultant correlation coefficient between the two-time series was 0.625 for the past 500 ka . We were then able to match DSDP 610B to the chronology of DSDP 610A using both XRF records.

To further constrain the age model for MIS 11 in DSDP 610B, we measured stable oxygen isotopes on *Uvigerina (spp.)* across MIS 11 from Termination V (TV) to MIS 11a. We tuned our benthic $\delta^{18}O$ record to the benthic $\delta^{18}O$ record of the well-dated Ocean Drilling Project (ODP) site 980 (McManus et al., 1999)

and its LR04 chronology, which has an uncertainty of ±4 ka (Lisiecki and Raymo, 2005) using AnalySeries (v. 2.0) (Paillard et al., 1996). The correlation coefficient between both records is 0.681. The regional benthic $\delta^{18}O$ record from ODP 980 was chosen over the conventional global LR04 benthic $\delta^{18}O$ stack record due to its close geographical proximity to our site. Based on this rationale, the sites should experience similar benthic $\delta^{18}O$ evolutions through time. In AnalySeries we adopted eight calibration points at 372 ka, 382 ka, 385 ka, 389 ka, 391 ka, 394 ka, 421 and 427 ka between DSDP 610B and ODP 980 based on both benthic $\delta^{18}O$ records (Fig. 3). The resulting age model for DSDP 610B affords accumulation of one cm of sediment every ca. 55 years. As secondary tie points, after adopting the benthic $\delta^{18}O$ derived age control points, we also compared the IRD records from ODP 980 (McManus et al., 1999; Oppo et al., 1998) and DSDP 610B which both show terrigenous inputs of similar magnitude and duration at ~390 ka during MIS 11b. The error associated with the duration of events and offsets was calculated by taking the standard deviation of the sedimentation rates per 0.5 cm between MIS 11b and 11c multiplied by the physical distance (in cm) between samples.

Following the age model approach used by Govin et al. (2012), we compared peaks in atmospheric methane recorded in the EPICA Dome C ice core record to our chronology for DSDP 610B ; adopting the same assumption that abrupt warming events are synchronous with warming across the wider North Atlantic region (Austin and Hibbert, 2012; Barker et al., 2015; Hodell et al., 2013). We note that the recovery from TV and the event recorded at ca. 390 ka in the Log (Ti/Ca) record occur very close in timing to the methane release following TV and the drawdown seen from ~386.5 to 390 ka (Loulergue et al., 2008; Nehrbass-Ahles et al., 2020) suggesting that the age model over MIS 11 for DSDP 610B is also consistent with the AICC2012 (Antarctic Ice Core Chronology 2012 – without additional tuning) to within hundreds of years (Fig. 4). The 390 ka methane event also corresponds to an enrichment of oxygen isotope values recorded in ODP 980 and 983 (EDC3 Age model (Barker et al., 2015)) and to a rapid cooling event (Uk'37-SST) recorded further south off the Iberian continental margin (Oliveira et al., 2016) (International Ocean Drilling Project (IODP) Site U1385). Based on this agreement we chose to plot data from ODP 983 according to the AICC2012 chronology as in Barker et al. (2015) and Barker et al. (2019) in favor over the LR04 chronology from 2004 (Raymo et al., 2004). For the time interval of interest, the two chronologies (e.g., AICC2012 and LR04) for ODP 983 are well within the ±4 ka uncertainty. However, this dating uncertainty does affect the certainty with which the relative timing of events described in this manuscript can be interpreted. Notably, the age models used here result in an excellent alignment of the benthic $\delta^{18}O$ enrichment event at 390 ka (Figure 3) which is a distinct feature at all sites (DSDP 610, ODP 980 and 983). Oliveira et al. (2016) denoted the event at 390 ka as isotopic event MIS 11.24 following the decimal event

notation of Bassinot et al. (1994), which further divides MIS 11b into a single light isotopic event, 11.23, conjoined on either side by the heavy isotopic events, 11.24 and 11.22.

High sedimentation rates (13.5 cm/ka) in DSDP 610B allowed us to reconstruct the evolution of climate from full interglacial conditions during MIS 11c (403 ka) to isotopic stage MIS 11c (390 ka) (Bassinot et al., 1994; Oliveira et al., 2016). According to our age model, MIS 11b thus takes place between 388.5-391 ka, which constrains the cooling event from onset to recovery to 2.0–2.5 ka. We compare our results to previously published datasets from IODP 303- U1308, and ODP 983 and 980 (Fig. 1), to provide regional context for palaeoceanographic interpretation of the observed events. We posit that the good correlation (structure and duration) between the three benthic $\delta^{18}O_c$ records from DSDP 610B, and ODP 980 and 983 during and leading up to MIS 11b allows a geospatial and relative temporal comparison of their respective climate archives. Acknowledging that direct comparison of age models is complicated by their respective uncertainties, all records discussed here (EPICA dome C, Red Sea core KL09, IODP 303-U1308, ODP 983, 980, and DSDP 610B) are nonetheless coeval with the Antarctic Ice Core Chronology AICC2012 age model (see Figs. 3-4) within uncertainties.

## 4.0 Results

Between 403 and 400 ka SST reconstructions provide evidence for a period of persistent warmth (e.g., 10-11°C), followed by a gradual increase in SST reaching 13-14°C by 397.5 ± 0.25 ka. Over the same period, foraminifer assemblages from the eastern SPG show an increase of transitional species, such as *Globigerina inflata*, a North Atlantic Current (NAC) and Atlantic Water indicator species (Kucera, 2007) (Fig. 5). At depth, we record a gradual increase in WTOW transport starting at 403 ka, reaching maximum values at 397 ka (Fig. 6).

Starting at 397 ka, however, we observe a total of two climate events over the interglacial to glacial transition. The first event at 397 ka begins with a decrease in WTOW flow speeds (397.43 ± 0.25 ka) over a ~710 ± 90-year period (Fig. 6; for estimation of uncertainties linked to the duration of events and offsets see section 2.11). After the initial decrease, WTOW remains weak but stable over ca. 5,360 years. The decrease in WTOW is followed by a two-step sea surface cooling 320 ± 90 years later. The two cooling steps are separated by a 2000-year plateau. The first cooling of 5°C occurred at 397.11 ± 0.26 ka over ~820 ± 50 years and the second cooling of 2°C took place over 250 ± 10 years from 394.45 ± 0.12 ka to 394.20 ± 0.13 ka. SSTs remain low for 960 ± 210 years and both cooling steps are accompanied by an increase in IRD of 100 grains.g$^{-1}$ and 1155 grains.g$^{-1}$ respectively (Fig. 6).

We note that the initial decrease in WTOW flow and subsequent cooling at the surface is separated by nine samples (4.5 cm), thus demonstrating that the observed offset of 320 ± 90 years is marked by a significant offset in core depth and indicates an actual delay between deep circulation and surface cooling. The recovery following the cooling at 394.45 ± 0.12 ka spans 2200 years, from 393.24 ± 0.26 ka to 391.06 ± 0.18 ka, during which SSTs increased slowly from 6°C to ~15°C. WTOW transport recovery to full interglacial values began between 391.74 ka and 391.00 ka. A more precise onset of the recovery is difficult to determine as a core break and some missing samples at exactly this location prohibit the calculation of statistically meaningful offsets (Fig. A2). Nevertheless, the onset of the recovery of WTOW with respect to SSTs was delayed by ca. 1500 years and occurred rapidly, over (at most) 740 ± 170 years. Foraminifer assemblages during the recovery see an increase in the dominance of transitional species, such as *G. inflata*, which reach maximum values of 31% at 390.4 ka reflecting an increase in SSTs from 14°C to 18°C near the eastern margin of the SPG (e.g., Rockall Trough) persisting for ca. 1160 ± 430 years. Meanwhile, at depth WTOW began to decrease again, reaching minimum values over 775 years from 390.50 ± 0.20 to 389.73 ± 0.25 ka.

At ca. 390 ka, our surface record shows the beginning of the second sharp, two-step cooling event, separated by a 360 ± 120-year plateau. The temperature decrease at the surface occurs 600 ± 260 years after WTOW flow decreases. It abruptly terminates the period of moderate surface climate. The first cooling step describes a rapid decrease of 5°C recorded by five samples over 2.5 cm. The second cooling step of 6°C occurs within six samples over 3 cm and lasts 220 ± 30 years. The duration of these events should be viewed as a maximum estimate given that bioturbation tends to smooth the signature of abrupt climate events in sediment cores (Anderson, 2001). Concurrent with the SST drop at 390 ka, IRD increases rapidly from zero to a maximum of 983 grains.gr$^{-1}$ peaking at ca. 389.5 ka. Comparatively, the climatic recovery from this second event as recorded in DSDP 610B was rapid and occurred over 400 ± 140 years, starting at 389.2 ± 0.07 ka. SSTs rose to ca. 12°C and varied thereafter between 11°C and 15°C and WTOW increase to reach initial interglacial values. Since WTOW values do not stabilize by the end of the timeseries produced for this study the recovery may extend beyond 388.5 ka. Like the previous recovery, surface warming precedes the increase in WTOW, however, the delay between the two is shorter covering only 300 ± 110 years. The observed increase in surface temperature and increase in WTOW flow is separated by 8 samples over 5.5 cm.

Thus, the waning stages of MIS 11 are marked by repeated, high magnitude, and rapid shifts in both deep overflow and surface climate. While largely coincident, centennial-scale phase lags at the onset and end of these anomalies do occur such that a decline in deep water flow precedes local climate cooling and

postdates the warming. In short, the climate amelioration falls clearly within the longer duration episode of reduced deep water flow.

## 5.0 Discussion

In the following discussion, we put our findings into the wider geographical context of the North Atlantic region to determine the processes and mechanisms that led to the described observations in WTOW and the surface ocean. We note that we examine the WTOW as representing the eastern Nordic Seas overflow more broadly. First, we discuss potential processes influencing WTOW over the described transition; we then analyse the two climate events at ca. 397 and 390 ka in detail.

### 5.1 WTOW and the wider geographic impact of a reduction in NDW.

Although our WTOW record only monitors one pathway of dense-water overflow from the Nordic Seas, comparison to other records suggests the changes we observe appear to be part of broader/general fluctuations in NDW influence. For example, benthic foraminifera stable isotope $\delta^{13}C$ records from nearby site, ODP site 980, located on the Feni Ridge (McManus et al., 1999; Oppo et al., 1998), as well as from sites further afield such as IODP site 303-U1308, ODP site 1063, and IODP site 303-U1385 monitoring lower North Atlantic Deep Water in the open Atlantic out of the direct path of the overflows from the Norwegian-Greenland Seas (Hodell et al., 2008; Nehrbass-Ahles et al., 2020; Poli et al., 2000), also show a reduction in deep Atlantic ventilation as early as 398.4 ka and 399 ka, respectively (Fig. 5 and 6). Thus, both dense overflow (WTOW) and broader ventilation proxies depict a weakening of NDW during these events.

We note that weakened WTOW flow and high-magnitude SST fluctuations invoke processes typically associated with millennial scale Dansgaard-Oeschger (D-O) events or Heinrich stadials (Denton et al., 2010; van Kreveld et al., 2000) when climate coolings are associated with weakened influence and export of NADW or NDW (Curry and Oppo, 1997; Elliot et al., 2002; Moros et al., 1997; Oppo and Lehman, 1995). Further, our results also align with previous research showing multi-century offsets between NDW production and abrupt surface climate changes during millennial-scale (D-O) events Robinson et al., 2017. Referring specifically to these temporal offsets in the response of NDW to surface climate forcing, our WTOW record, therefore, suggests that mechanisms previously observed only during intermediate or large cryosphere boundary conditions also appear to operate for climates characterized by low ice volume conditions.

In investigating triggers for D-O cycles of MIS 3, Dokken et al. (2013) proposed a temperature threshold in the Nordic Seas that, once passed, causes widespread sea ice coverage over deep water ventilation areas, resulting in a weakened deep circulation and strong surface cooling in the North Atlantic (Dokken et al., 2013). In the context of a glacial inception, a temperature threshold in the Nordic Seas has also been invoked to explain rapid North Atlantic cooling at the onset of the last glacial inception (Born et al., 2010). Given the progression into a Northern Hemisphere summer insolation minimum (June) and likely cooling of high northern latitudes at 397 ka, the possibility of cooling over the Nordic Seas concurrent with an enhanced export of sea ice and freshwater to deep water formation sites is plausible. Such an insolation configuration would tend to drive the ocean toward an equilibrium state with reduced NDW influence relative to Antarctic Bottom Water (Galbraith and de Lavergne, 2019). Further, Yin et al. (2021) most recently hypothesize that abrupt weakening of the AMOC at the end of interglacial periods (but before glacial boundary conditions are established) could be triggered by a combination of a high precession with June solstice occurring at aphelion and, at the same time, a relatively low obliquity (inducing low total summer irradiation). This hypothesis is consistent with our proxy observations at 397ka when precession is high and obliquity low (entering minima), (e.g., Yin et al., 2021). Like Born et al. (2010); Yin et al. (2021) also suggest a high latitude sea-ice feedback mechanism to explain the sharp decline in overturning terminating interglacial warmth. In support of a high latitude trigger for the decrease in NDW formation, our data show that decreasing WTOW preceded the cooling of SSTs and the occurrence of IRD at subpolar latitudes. In fact, for both instances of cooling described here, changes in overflow preceded surface cooling over the Rockall Trough by centuries (320 and 710 years, respectively). This suggests that local influences such as the properties and routing of subtropical waters on their northward trajectory were not the immediate cause of changes in deep overflow (locally, or more broadly) but responded late to reduced overflows and had to recover before overflow increased.

## 5.2 Reorganization of Atlantic Waters in the SPG

In the wider palaeoceanographic context, the onset of sea surface cooling observed at our site over the Rockall Trough at ca. 397 ka also occurs at site M23414 200 km west of DSDP site 610 (Fig 6b; Kandiano and Bauch (2007)) but is not evident further west, closer to the SPG (ODP site 983, Gardar Drift 60.48 N, 23.68 W), where *Neogloboquadrina pachyderma* abundances remain low and stable (0–10%) (see also Fig. 6) until ca. 391 ka (Barker et al., 2015). While we cannot strictly rule out that the 391ka cooling observed at ODP 983 is actually occurring at 397 ka, most age models published for this site (EDC3, AIC2012; Barker et al. (2015)) place this event around 391 ka (except LR04; Raymo et al. (2004) which places it closer to 393 ka) and this timing is supported by the close alignment of the distinct benthic $\delta^{18}$O excursion at all North Atlantic sites (Figure 3; cf chronology section) and the methane changes at this time

 (Figure 4). In the absence of changes at the edge of the SPG (ODP 983), it is unlikely that a major displacement of oceanic fronts, as found by Irvalı et al. (2016) and Mokeddem et al. (2014) to mark the demise of the last interglacial, occurred at this time. Instead, we propose that the cooling observed around 397 ka at DSDP 610 and M23414 resulted from a more complex redistribution of subpolar surface waters in the North Atlantic region including adjustments within the main northward-flowing branch of the Thermohaline circulation. Potentially, these changes invoked a shift in the overall configuration of the SPG from a North-South to an East-West orientation, as described for the Holocene by Thornalley et al. (2009). Alternatively, cooling of NAC source waters may have resulted in the SST decrease at DSDP 610. However, if tropical SSTs evolved similarly as observed over the mid-to-late Holocene transition, the transition into an insolation minimum at high northern latitudes and an insolation maximum at low latitudes would have favoured warmer, rather than cooler, SSTs in tropical source regions (Santos et al., 2013).

Unlike the first sea-surface cooling observed at 397 ka, the onset of cooling at 390 ka at DSDP 610B is preceded by an abrupt cooling at the eastern edge of the subpolar gyre at 390.8 ka (Barker et al., 2015). Barker et al. (2015) previously interpreted this event as reflecting a gradual regional cooling corresponding to the southward migration of fronts, where warm Atlantic Waters inflow into the Nordic Seas is maintained even if the Polar Front had moved south of ODP site 983 (Barker et al., 2015). In support of this interpretation, we note that the advance of the Sub-Arctic Front over ODP site 983 appears synchronous with an increase in SSTs from 14°C to 18 ± 1.8°C near the eastern margin of the SPG (e.g., Rockall Trough), which persisted for ca. 1160 ± 430 years. An increase of SST peaking at 390.47 ka is also evident in the SST record from M23414 (Kandiano and Bauch, 2007), although the low resolution of this record (ca. 400 years) complicates the detailed comparison of this sea surface warming between sites. We infer that the observed SST increase describes this enhanced inflow of Atlantic Waters at a time of reorganization in oceanic fronts and the SPG structure, potentially attributable to gyre weakening.

This interval of maximum SST in our record was followed by an increase from 4% to 12% of Arctic Front indicator species *Turborotalita quinqueloba*, suggesting that this phase describes the gradual approach of the Sub-Arctic Front, as hypothesized previously for precursor events to the glacial inception at the end of MIS 5e (Irvalı et al., 2016; Mokeddem et al., 2014) (see Fig. 5). Assemblages at M23414 are too low in resolution to assess these multidecadal to centennial-scale observations, however, there is generally good agreement between the two *T. quinqueloba* records (Fig. 5). The stepwise increase in abundance of polar foraminifera *N. pachyderma*, first from 20% to 50% at 389.90 ka ± 0.60 and again from 50% to 85% 360 years later, demonstrates the passage of oceanic fronts over the Rockall Trough (Johannessen et al., 1994) (Fig. 5). IRD also increases rapidly from zero to a maximum of 983 grains.gr$^{-1}$ following the passage of

the Arctic Front. We note that *N. pachyderma* assemblages at M23414 are also increasing to 28% and IRD counts increase to 348 grains.gr$^{-1}$ until 389.09 ka, which is the last datapoint in this time series (Kandiano and Bauch, 2007).

The movement of oceanic fronts and the vast reorganization of Atlantic Waters at subpolar latitudes at 390 ka coincides with a peak in summer insolation at 65°N (Ganopolski et al., 2016). Modelling simulations link increases in boreal summer insolation at high northern latitudes with a strong summer melt season, elevated SSTs in the Nordic Seas, and reduced sea-ice extent (Tuenter et al., 2005). Specifically, the event at 390 ka occurs when precession is low (Northern Hemisphere summer at perihelion), and obliquity is low (exiting minima), which invoke processes described in Zhang et al. (2021) who propose that AMOC variability can be triggered by either precession or obliquity during intermediate cryosphere climate background. The combination of low obliquity and high summer insolation at high latitudes would have led to intense summer heating of the emergent GIS and Scandinavian Ice sheet, which would have led to a warm Nordic Seas and set the stage for intense melting at 390 ka. Further, we note that greenhouse gases at the time (values of $CO_2$: 259.5 ppm and $CH_4$: 568 ppb) were still at or close to interglacial values (Fig. 6), amplifying radiative warming further (Bereiter et al., 2015; Petit et al., 1999). We speculate that the combination of high insolation and greenhouse gas radiative forcing caused the described oceanographic changes at subpolar latitudes. Further evidence for significant melting of the cryosphere at 390 ka is also provided by a distinct reversal of RSL lowering over the event (Grant et al., 2014; Spratt and Lisiecki, 2016).

**5.3 Mechanistic link between climate boundary condition and Deepwater response.**

We observe a reduced NDW influence concurrent with a freshening of the central SPG (Fig. 6; (Barker et al., 2015)). This reduction of the eastern overflows is also seen in benthic $\delta^{13}$C records from across the North Atlantic SPG (Galaasen et al., 2020; Hodell et al., 2008; Nehrbass-Ahles et al., 2020; Oppo et al., 1998; Poli et al., 2000) suggesting that both Nordic Seas overflows were impacted at this time, but the event is less clearly defined in records from the deep South Atlantic (e.g., ODP Site-1085, 1713m water depth and MD07-3077, 3400m water depth Dickson et al., 2008; Riveiros et al., 2013). How such a deep water weakening arises and persists remains unclear but has clear climate consequences as evidenced in our data. We offer two possible mechanisms that can account for this episode of reduced NDW influence. First, freshwater forcing to the Atlantic Ocean basin from melting of (proto) circum-North Atlantic ice sheets could potentially stratify the surface ocean in the Nordic Seas and central SPG, decreasing sea-surface salinity (Dokken et al., 2013; van Kreveld et al., 2000). In response, the AMOC undergoes a freshwater-induced transition to a circulation mode where NDW formation is significantly decreased

(Ganopolski and Rahmstorf, 2001). This weakened deep water state is maintained through internal processes including air-sea interactions and feedbacks on ocean advection where freshening in the SPG causes a reduction in the northward transport of warm high-salinity waters (Kostov et al., 2019). Simulations offer a range of plausible feedback mechanisms operating in the coupled atmosphere-ocean-sea ice system for amplifying an initial perturbation (Drijfhout et al., 2013; Kleppin et al., 2015; Li and Born, 2019; Rind et al., 2018). For example, in response to freshwater input, a stronger advection of warm air from North America and the North Atlantic to the Labrador and Nordic Seas occurs causing a reduction in sea surface heat flux (Justino and Machado, 2010). This reduces the air-sea temperature difference, which also reduces the amount of heat lost from the ocean to the atmosphere leading to shallow convective mixing and weaker NDW formation (Justino and Machado, 2010). More generally surface buoyancy fluxes and specifically air-sea feedback mechanisms, such as changes in evaporation and precipitation in response to temperature and aerosol feedbacks may ultimately amplify and maintain an initial reduction in deep water production in response to freshening (Rind et al., 2018).

Regardless of the specific processes and feedbacks giving rise to persisting interglacial circulation anomalies, their existence indicates that non-linear behavior in the climate-ocean system is not strictly confined to glacial intervals. To explain glacial millennial-scale variability Menviel et al. (2020) proposed a self-sustained mode of the coupled climate-ice sheet system, whereby centennial-scale changes in freshwater balance (increased meltwater run-off from circum-Atlantic ice sheets); decreases in $CO_2$ concentration; and/or changes in North Atlantic wind stress could lead to AMOC weakening, associated sea-ice advance (Jensen et al., 2018), and a southward shift of deep water formation sites (Lynch-Stieglitz et al., 2006). Models are increasingly able to generate such bifurcations in ocean-atmosphere circulation, and thus climate, and an increasing number of models produce these transitions spontaneously (Brown and Galbraith, 2016; Vettoretti and Peltier, 2015). Yet, many of these models and mechanisms are invoked to explain climate and circulation transitions during glacial episodes (e.g., for D-O variability) and it is not clear to what extent these mode changes and mechanisms are relevant to interglacial (high CO2, low ice volume) states nor what the prerequisites are for activating them. However, sea ice-atmosphere-ocean coupling has also been found to give rise to spontaneous AMOC transitions in model simulations of previous interglacial periods suggesting that large ice sheets may not be required to sustain ocean circulation anomalies (at least for many centuries; (Kessler et al., 2020)).

## 5.4 Role of cryosphere

While the GIS is thought to have been considerably reduced during MIS 11c (Robinson et al., 2017), the presence of IRD in the eastern SPG at ca. 397 ka requires marine-terminating glaciers or ice sheets before

the onset of the first cooling event at 397 ka. Evidence for ice sheet growth prior to ca. 397 ka comes from benthic stable oxygen isotope ($\delta^{18}O$) data and relative sea-level (RSL) curve reconstructions indicating a slow build-up of ice sheets beginning at ca. 400 ka (Fig. 6), with -1.02 ± 24 m RSL by 397 ka, according to the RSL curve from Spratt and Lisiecki (2016) (see also: Bailey et al., 2012; Hodell et al., 2008; Robinson et al., 2017). The presence of IRD is not necessarily synonymous with an increase in the production of icebergs at that time but may instead indicate increased iceberg survival to the core site, made possible by the southward expansion of cool surface conditions, enhanced atmospheric circulation, and cooling of the central SPG at this time (Bailey et al., 2012; Hodell et al., 2008; Irvalı et al., 2020). Over the inception we record three distinct IRD events at 397, 393, and 390 ka of 100, 1155, and 983 grains.$g^{-1}$ respectively. These events occurred when global ice volumes were -1.02 ± 24 (397 ka), -9.74 ± 24 (393 ka), and -28.4 ± 13 RSL (390 ka) (Spratt and Lisiecki, 2016). Each of these events, including the smallest at 397 ka which occurred when sea levels were comparable to pre-industrial times, exceed Holocene IRD variability of 0-20 grains.$g^{-1}$, assessed from nearby core VM29-191 (Bond et al., 1999), and in the case of IRD at 393 and 390 ka, these events are more akin to counts observed during the Younger Dryas (800 grains.$g^{-1}$) (Bond et al., 1999). In this context and in an attempt to determine the source of IRD at DSDP site 610, it is important to note that IRD at 393 and 390 ka has also been detected within two distinct peaks in detrital silicate (via XRF (Si/Sr)) at IODP Site 303-U1302 (Fig. 1) located off the Newfoundland continental margin in the western Atlantic basin (Channell et al., 2012). The two peaks are coeval with the IRD peaks in DSDP 610B within dating uncertainties (Fig. A3), supporting a northern or north-western source for IRD (e.g., GIS or LIS) rather than a surge from the east (e.g., British Irish Ice Sheet or BIIS). Such a source for IRD would also explain the absence of IRD near IODP site 983. The peak in detrital carbonate (Ca/Sr) in IODP site 303-U1302 at 394 ka further supports this argument and may indicate at least a partial contribution of IRD from the Laurentide Ice Sheet to the IRD recorded at DSDP site 610 at this time. At site IODP site 303-U1308, south of DSDP site 610, the evidence for IRD is muted at 390 ka and absent at 394 ka (as recorded in the Si/Sr record) (Hodell et al., 2008). Together the evidence from IODP sites 303-U1302, 303-U1308, M23414, and DSDP 610 would therefore support a Northern/Western origin of IRD rather than IRD from the BIIS, however further analysis would be needed to confirm this hypothesis. Crucially, the magnitude of WTOW flow reduction associated with even the smallest IRD event at 397 ka is the same as for the larger IRD event at 390 ka (Fig. 6 and Fig. A2).

The geographic impact of meltwater in the surface ocean appears to be commensurate with the magnitude of IRD recorded at our site. At 397 ka changes in surface ocean circulation appear to have been restricted to subpolar latitudes. While, at 390 ka IRD and surface water cooling were observed as far south as the Iberian Margin in conjunction with a reversal of sea levels evident in several RSL reconstructions (Grant

et al., 2014; Oliveira et al., 2016; Rodrigues et al., 2011; Spratt and Lisiecki, 2016). Though we recognize that the magnitude of IRD is not linearly related to the magnitude of iceberg calving, the similar response of WTOW accompanying IRD events of varying magnitudes all suggests that perhaps the likelihood of these events is not linked to a threshold in ice volume, but rather to the rate and volume of freshening occurring regardless of boundary conditions. It is important to note that while SST records outside the SPG do not record climate variability between 397 and the event at 390 ka, pollen reconstructions from the Iberian margin show evidence for two rapid climate events centered at 396 ka and 393.5 ka (Oliveira et al., 2016) concurrent with the first two-step cooling event described here (within dating uncertainties). If indeed linked, the pollen record would potentially suggest an atmospheric coupling with the reorganization of Atlantic Waters at subpolar latitudes, linked to changes in mid-latitude climate.

### 5.5 Climate recovery

The recovery following the second cooling event (ca. 390 ka) was rapid, taking only 400 years for SSTs to reach 12°C. The onset of WTOW recovery lags SST increase by $300 \pm 110$ years, which is significantly shorter than our observations at 393 ka. However, it is interesting and potentially important to note that in both instances WTOW only recovered after SSTs over the Rockall Trough reached ca. 9°C. At ODP site 983, sea surface warming was similarly delayed by ca. 400 years and occurred more gradually. We propose that this paleogeographic pattern of the climatic recovery in the surface ocean reflects the location of ODP site 983, the relative positioning of oceanic fronts relative to the core site, and the localized or regional impact of meltwater from ice sheets on SSTs. These observations outline a rapid east-to-west retreat of Polar Waters during the recovery allowing warm Atlantic Waters to return to the eastern North Atlantic.

That the recovery from MIS 11b was part of a broader, potentially global climate event is indicated by the concurrent and pronounced increase in Antarctic temperature (Jouzel et al., 2007), global sea level (Grant et al., 2014; Spratt and Lisiecki, 2016), and atmospheric concentrations of $CO_2$ and $CH_4$ (Nehrbass-Ahles et al., 2020), which together suggest that the recovery after 390 ka involved global scale teleconnections between ocean circulation, atmospheric temperature, and ice volume (Fig. 6). A potential amplifying mechanism, that could add to the abrupt (local) high magnitude SST rise (10-12°C) during the recovery, and the resumption of WTOW within a global context, was described by Ballalai et al. (2019). Their palaeoceanographic investigation provides evidence for the development and subsequent release of warm and saline Atlantic Waters from the South Atlantic into the North Atlantic Basin, in conjunction with a northward shift of the Intertropical Convergence Zone (ITCZ) at 129 ka (MIS 5e). At both times, 129 ka (MIS 5e) and 390 ka (MIS 11b), annual insolation at 65°N was increasing, providing similar antecedents for a sudden release of warm and saline South Atlantic Waters into the North Atlantic and the resumption

of deep overturning in the Nordic Seas. A sudden northwards shift of the ITCZ, would also support mechanisms leading to a sudden release of $CO_2$ and $CH_4$ (e.g., formation of new tropical wetlands, permafrost thawing, etc., in the northern hemisphere) during the recovery after 390ka (Nehrbass-Ahles et al., 2020). While speculative, this mechanism would agree with foraminifer assemblage compositions from DSDP 610B, where we see an increase in *G. bulloides* to 15.5%, suggesting a stronger inflow of warm water via the NAC, and an increasing dominance of transitional species, including *G. inflata* that reaches maximum values of 31% at 390.4 ka. Although the trigger may be extra-basinal, generating such large anomalies in Atlantic Water inflow pathways (e.g. 10 °C) might also require intrabasinal changes (e.g. in SPG and Atlantic inflow geometries) in response to the large atmospheric shifts (Ballalai et al., 2019) thought to accompany these anomalies.

**6.0 Conclusion**

In summary, the palaeoceanographic signature of the first cooling event at ca. 397 ka appears to share similarities with the last glacial inception that occurred 119–115 ka (Born et al., 2011; Mokeddem et al., 2014). We suggest that the more gradual nature of this transition, both in terms of surface cooling and deep-water flow reduction, is symptomatic of the gradual but low-volume increase in sea ice and freshwater export to NDW formation region and to subpolar latitudes linked to decreasing summer insolation at that time. In contrast, the considerably more abrupt climate response and reorganization of Atlantic Waters in the SPG during MIS 11b is more characteristic of a meltwater event, in this instance co-modulated by elevated greenhouse gases and a prolonged summer melt seasons at high northern latitudes, where a significant build-up of land Ice (-30 m ± 20 RSL) had taken place during the preceding precession cycle.

Irrespective of the magnitude of climate variability or boundary conditions, the reorganization between Polar and Atlantic Waters at subpolar latitudes appears to influence deep water flow in the Nordic Seas. Our data show that a reduction in deep water flow precedes surface hydrographic changes in the eastern Atlantic during both glacial and interglacial boundary conditions. These observations provide evidence that similar processes and timescales, previously described only for abrupt climate events during the last glacial period (e.g., Muschitiello et al., 2019), also operated during interglacial and/or low-ice climate states.

The regulation of subpolar surface buoyancy must be central mechanistically, and we postulate that the duration and magnitude of these climate events might be modulated by the availability and/or rate of freshwater reaching NDW formation regions and subpolar latitudes perhaps initiated by GIS or sea ice melt but potentially strengthened through air-sea feedbacks (precipitation over evaporation) as simulated for

high $CO_2$ global warming scenarios (Rind et al., 2018). We hypothesize that it is the rate (Lohmann and Ditlevsen, 2021) and/or volume of freshwater which dictates the occurrence of these types of climate events (e.g., rate-induced tipping point), regardless of boundary conditions; and while the presence of larger ice volumes may facilitate the occurrence of abrupt climate events, pattern and phasing of their climate impacts during low ice volume periods appear to be similar. This is important in the context of modern and predicted future melting of the GIS and particularly pertinent given the general level of concern for AMOC sensitivity to future buoyancy changes and the fact that circulation in the North Atlantic already appears to be undergoing significant changes (Caesar et al., 2021; Rahmstorf et al., 2015; Smeed et al., 2014; Srokosz and Bryden, 2015; Thornalley et al., 2018).

**Data and materials availability:** All data needed to evaluate the conclusions in the paper are presented in the paper and/or the Supplementary Materials. Raw data will be made available in Pangaea upon publication.

**Author contributions:** The research and GSI proposal was designed and managed by A.M. in collaboration with S.T. and U.N.; D.H. performed faunal counts, sediment size analysis, IRD counts, data analysis, and wrote the first draft of the manuscript; U.N. performed stable isotope analysis; G.P. developed noise model and provided confidence intervals for endmember analysis; M.C. collected XRF data; D.H., A.M., G.B., U.N., G.P., and T.B. contributed to discussions and wrote the final version of the manuscript.

**Competing interests:** The authors declare no competing interests.

**Acknowledgements** This research was funded by the Geological Survey Ireland Short Calls Research Program awarded to A.M., award number 2017-sc-028. U.N. acknowledges establishment funding for the stable isotope facility FARLAB (Research Council of Norway award number 245907). D.H. acknowledges support from the Environmental Protection Agency and Irish Research Council under grant GOIPG/2020/14. G.P. is funded by a Natural Environmental Research Council Independent Fellowship (NE/P017266/1).

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

**Figures and Tables**

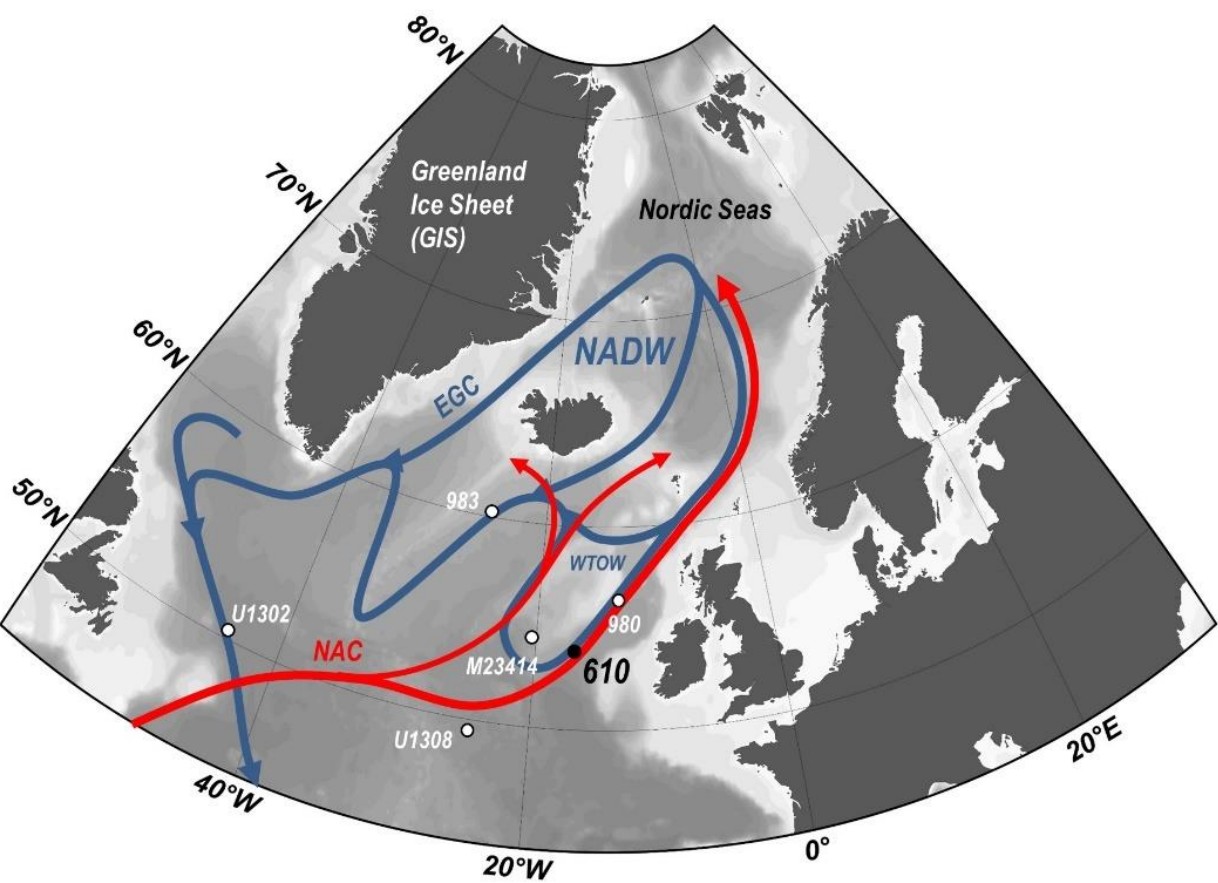

**Figure 1 | Regional context of the study area.** Schematic representation of the North Atlantic Ocean and Nordic Seas with arrows indicating the circulation components of the Atlantic Meridional Overturning Circulation (AMOC) in the North Atlantic basin. Major ocean currents include the North Atlantic Current (NAC) in red and in blue the deep water originating in the Nordic Seas (NDW) in blue. Also shown are the locations of the East Greenland Current (EGC), DSDP site 610 (this study), ODP site 980, ODP site 983, IODP sites 303-1302, 303-U1308 and site M23414. This figure was generated using Ocean Data View
software (http://odv.awi.de/).

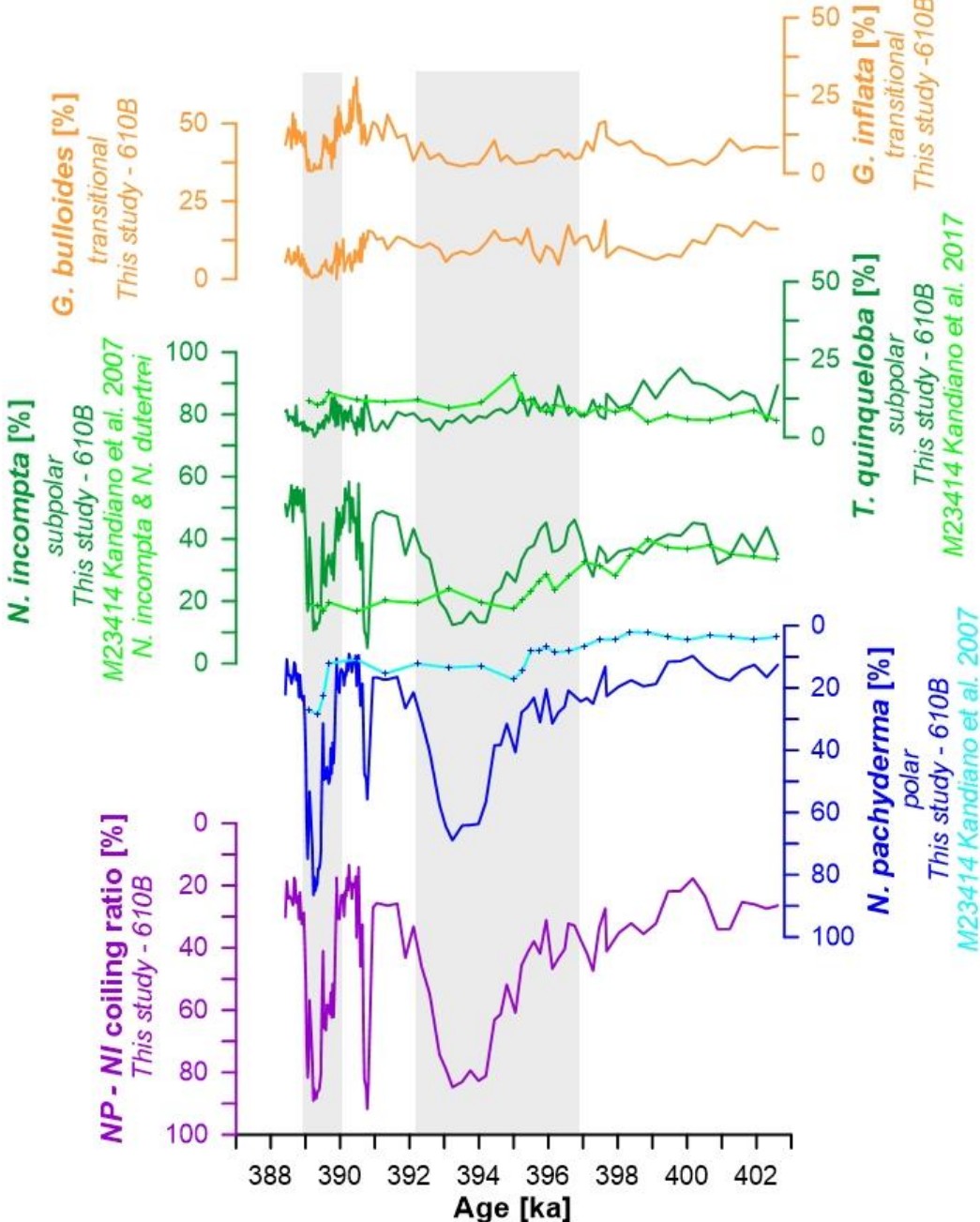

**Figure 5 | The evolution of planktic foraminifera over the MIS 11 glacial inception.** From top-to-bottom, the relative abundances of *G. inflata* (*transitional*, *orange*), *G. bulloides* (*transitional*, *orange*), *T. quinqueloba (subpolar, green)*, *N. incompta (subpolar, green)*, *N. pachyderma (note reversed axis, polar, blue)*, and coiling ratio (%) of N. pachyderma to *N. incompta* (*note reversed axis*, purple). Also shown are assemblage counts for *T. quinqueloba*, *N. incompta*, and *N. pachyderma* from nearby site M23414 (Kandiano et al. 2007).

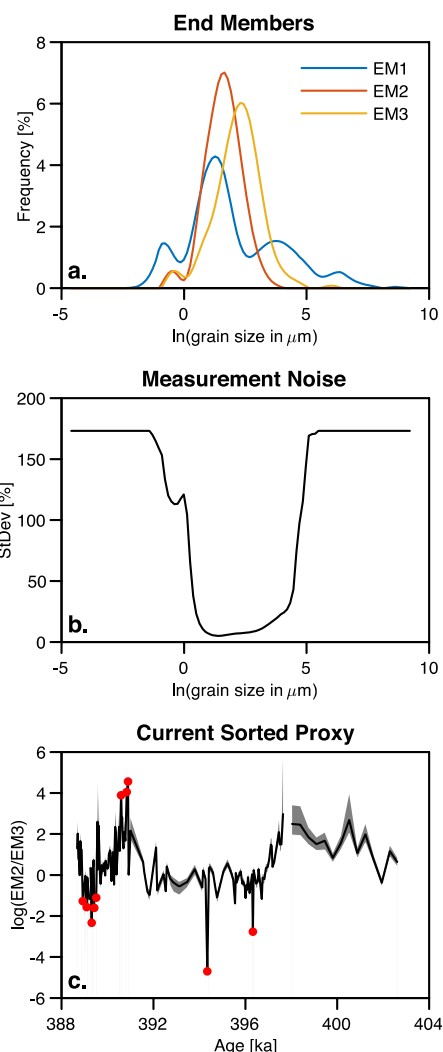

**Figure 2 | Fitted end members and data noise model for current-sorted proxy for DSDP 610B.** a) Grain size distributions for the three fitted end members. b) Estimated noise model for the grain size measurements. c) The current sorted proxy (ln(EM2/EM3)). The grey shaded area indicates the 95% confidence interval. Red dots denote where the confidence interval is undefined due to low abundances of EM2 and/or EM3.

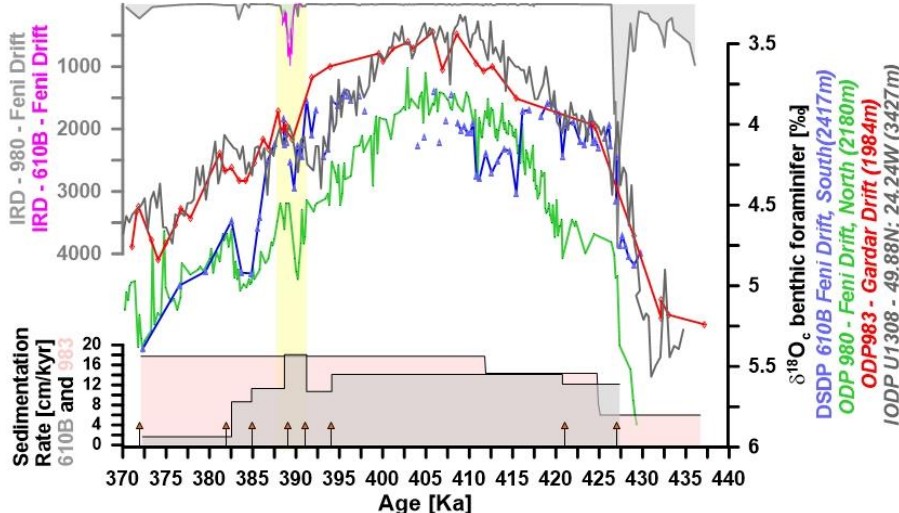

**Figure 3 | Age model development for DSDP 610B (370-470 ka).** Top-down: IRD records from DSDP 610B (pink) and ODP 980 (grey) from the Feni Drift; $\delta^{18}O$ benthic foraminifera records from DSDP 610B (blue), ODP980 (green, and ODP 983 (red), IODP 303-U1308 (brown) corrected by 0.63 ‰; sediment rate in cm/ka for DSDP 610B (grey) and. ODP 983 (light-pink). Yellow band marks the MIS11b event. Arrows indicate tie points.

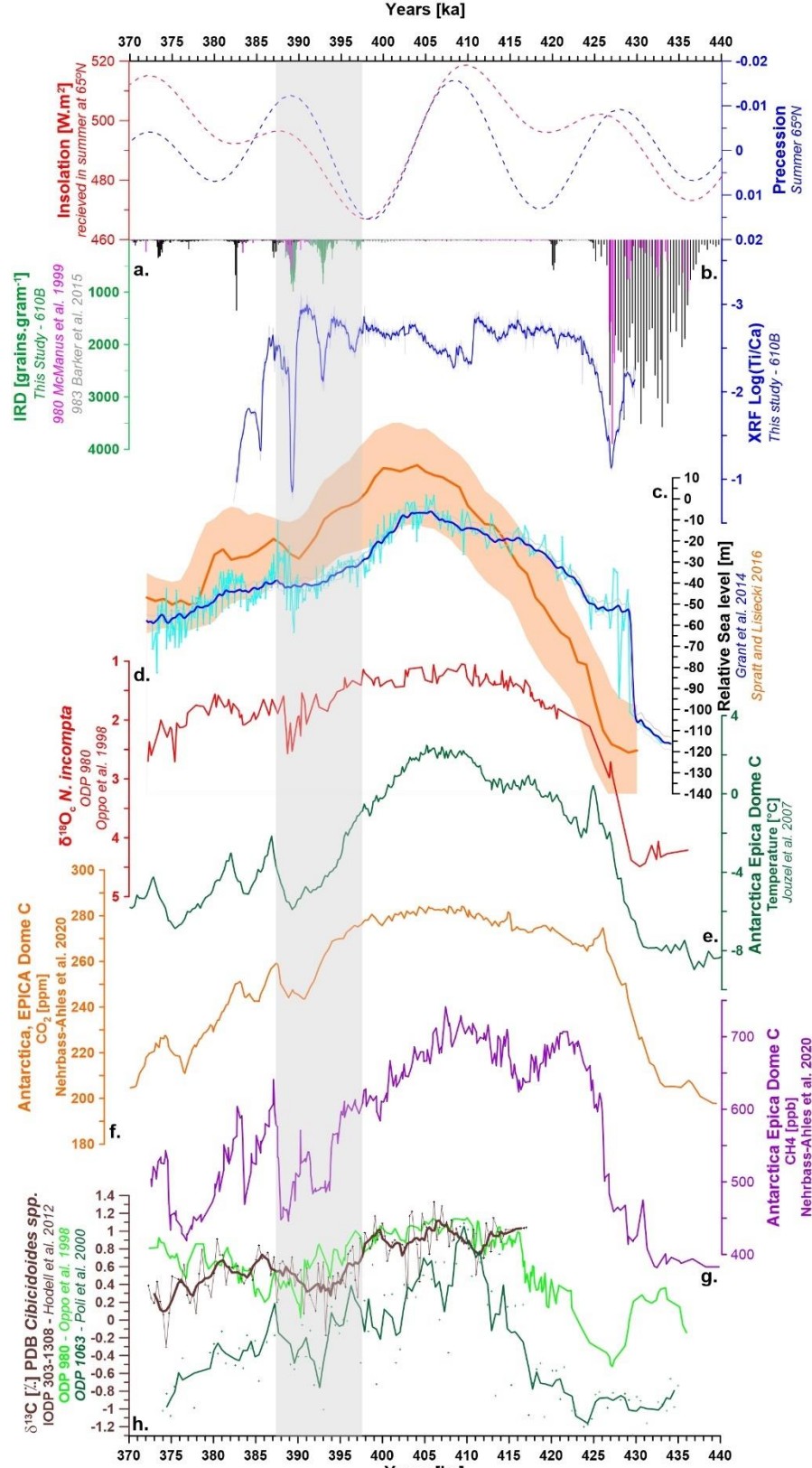

**Figure 4 | Age model development for DSDP 610B (370-430 ka ago). a**. IRD per gram records from DSDP 610B (green), ODP site 980 (pink), and ODP 983 (black); **b**. log (Ti/Ca) record for DSDP 610B (blue); **c**. global relative sea-level curve (blue and orange); **d**. ODP 980 δ¹⁸O *N. incompta* (red); **e**. EDC Antarctic ice core temperature (green); **f**. Carbon Dioxide (CO₂; orange); **g**. Methane records (CH₄; purple); and **h**. Benthic δ¹³C record from ODP 980, IODP 303-U1308, and ODP 1063 (brown light green and green). The grey band indicates the interval of this study.

1175

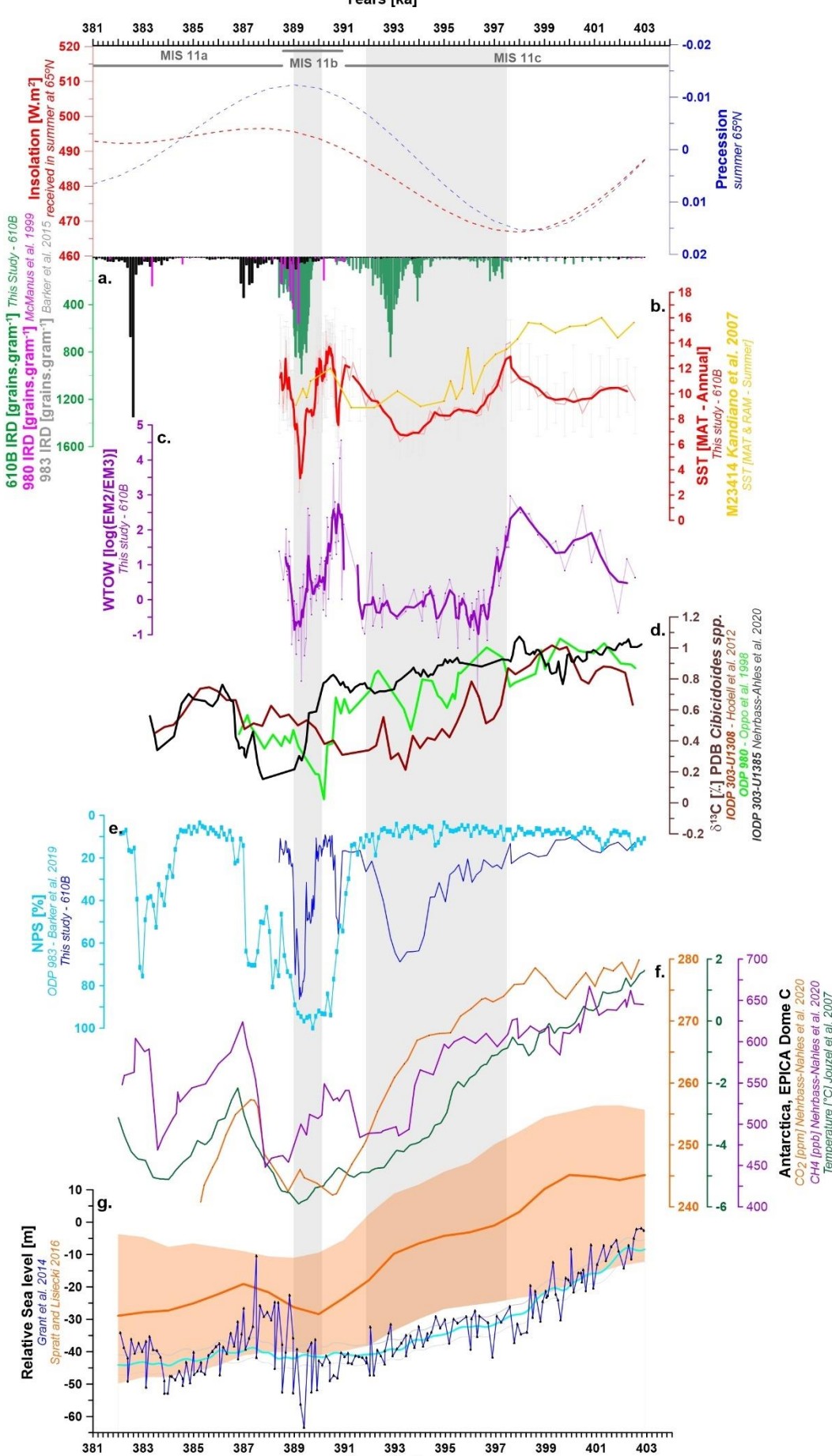

**Figure 6 | MIS 11c to 11b proxy records from DSDP 610B**. **a**. IRD per gram records from DSDP 610B (green), ODP 980 (pink), and ODP 983 (black); **b**. Summer SST estimates from MAT (red); **c**. The log ratio of EM2 and EM3 as a proxy of deep water flow strength (purple); **d**. $\delta^{13}$C records from ODP 980 (3pt-running average: light green), IODP 303-U1308 (3pt-running average: dark brown) and IODP 303-1385 (3pt-running average: black line; **e**. 983 NPS % record (light blue); **f**. EDC Antarctic ice core temperature (green), $CH_4$ (purple), and $CO_2$ (orange) records; **g**. Global relative sea-level curve (blue and orange). In parts b, c, and d, pale lines represent measured data and the bold lines are 3-point running averages.

## Appendix

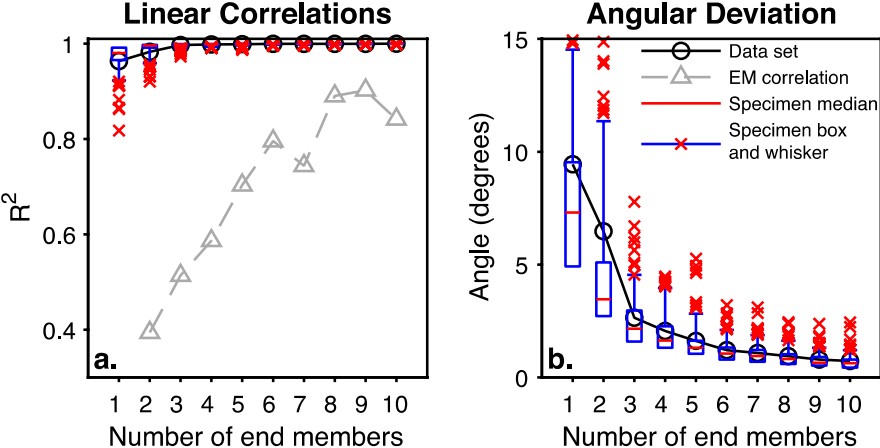

**Appendix Figure A1 | Goodness-of-fit assessment for non-parametric EMA.** a) $R^2$ and b) angular deviation ($\theta$) parameters as a function of the number of fitted end members. Both indicate that three end members are sufficient to describe this data set.

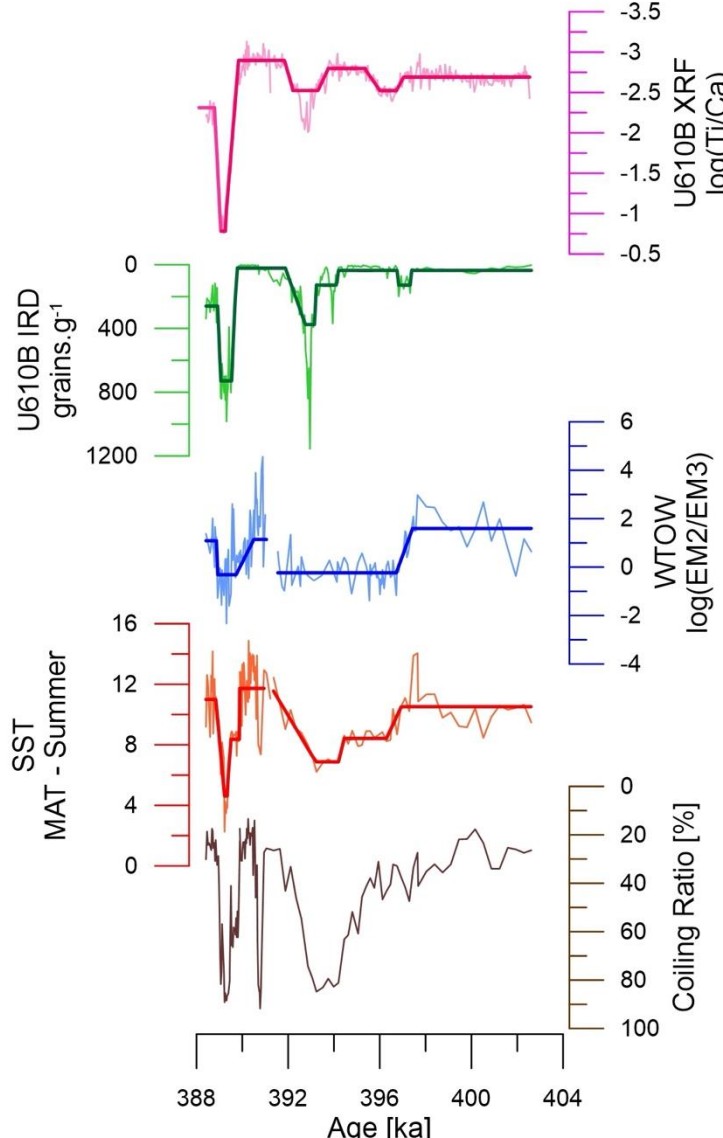

**Appendix Figure A2 | Ramp function fit (heavy line) to DSDP 610B time series (Appendix Table A1).** Time series (from top to bottom)
DSDP 610B XRF Ti/Ca (pink); IRD (green); WTOW proxy (blue); SST (red); and the coiling ratio of N. pachyderma to *N. incompta*
(brown).

**Appendix Table A1 | Analyzed time series.** Time units refer to thousand years before present. V is the coefficient of variation of the time
spacing (i.e., the standard deviation divided by the average).

| Name | Description | Time interval | n | V |
|------|-------------|---------------|---|---|
| DSDP 610B | SST | [388.421 ka; 402.612-ka] | 129 | 1.05 |
| DSDP 610B | WTOW | [388.420-ka; 402.610-ka] | 143 | 1.12 |
| DSDP 610B | IRD | [388.421-ka; 402.612-ka] | 211 | 1.19 |
| DSDP 610B | XRF | [388.421-ka; 402.612-ka] | 375 | 0.27 |

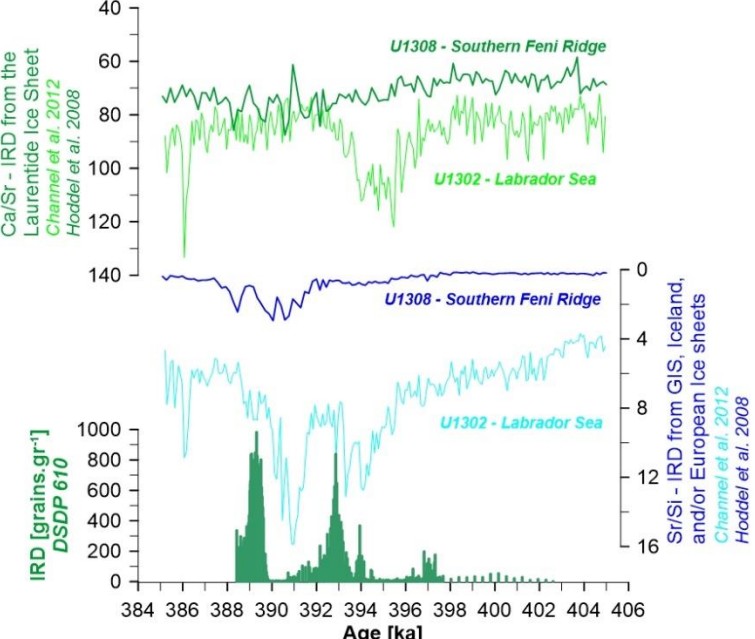

**Appendix Figure A3 | Assessment of IRD provenance.** X-ray fluorescent datasets from IODP 303-U1308 and IODP 303-U1308 are used next to IRD counts from DSDP 610B to evaluate the source of IRD for the glacial inception from MIS 11c into 11b. The strong evidence for the presence of detrital silicates (Sr/Si) in the western Atlantic Basin (IODP 303-U1302) support the hypothesis for a northern or western source of IRD as opposed to an eastern origin from the British-Irish Ice sheet.