# Peer review of "Reorganization of Atlantic Waters at sub-polar latitudes linked to deep water overflow in both glacial and interglacial climate states."

_Climate of the Past, 2021_

## Author Comment (AC2)

**Point by point response to Reviewer 1**

We would like to thank Reviewer 1 for helpful comments on our manuscript. Here we have addressed each of the comments and questions in the following format: Each question or comment is re-stated as in the original review of the manuscript in black 'Calibri font'. Our response to each comment/question is indented and written in blue 'Calibri font'.

Firstly, we feel that the way that the paper is set up does not really give a true reflection of the papers findings. The paper claims that abrupt climate events are;
1) a characteristic of glacial climate states, and
2) the results of this study (showing abrupt events occurred at the transition between MIS 11c and 11b) question this and our ideas about our concept of warm climate stability.
Neither of these claims are strictly true.

> We edited the abstract, introduction, and conclusions to reflect our findings more clearly.

It has long been known that abrupt climate events occur during interglacials – the 8.2 ka event (along with the pre-boreal oscillation, 4.2 ka event and 2.8 ka event) is a good example of this. Whilst it is likely that none of these were of a magnitude or duration comparable to the Lateglacial Interstadial (i.e. the Younger Dryas) they had transformative effects on ecosystems, surface processes and societies so are still significant. The concept of "warm" climate stability was surely abandoned a long time ago.

- First, we would like to clarify that the main objective in this manuscript is not to claim that our findings are the first to identify high magnitude climate events during interglacial boundary conditions, but instead, our goal is to elucidate their mechanistic links in the climate system specifically involving both surface and deep ocean circulation changes.
- Further, we believe that it is still the predominant paradigm that warm climates are less likely to experience high-magnitude (hemispheric/global) climate events because a large cryosphere is believed to be required as a key antecedent for high-magnitude events to occur and persist (e.g., Zhang et al. 2021). As a result, our understanding of how high-magnitude events did or may occur in the future without NH ice sheets is not well developed. Part of the rationale for the study presented here is that we do not yet appreciate the full scale of antecedent conditions that can lead to high magnitude climate instability.

If the authors are distinguishing between high magnitude (glacial) and low magnitude (interglacial) abrupt events then they need to do so more clearly and, ideally, in a quantified way.

> We agree that such a definition would be helpful. In the introduction, we now include a clarification that the maximum range in Holocene SST variability in the SPG does not exceed 3-4 °C (e.g., (Thornalley et al., 2009) while glacial DO events are characterized by SST changes of >6°C (Dokken et al., 2013).

Regardless of how abrupt events are defined, the study presented here doesn't move this argument forward. The events discussed here occurred on the climatic downturn into MIS 11b and, consequently, long after fully interglacial conditions had ceased. They are actually more true of abrupt events under a glacial, or transitional, state in that, as shown in Figure 6, a significant fall in sea level had already been experienced prior to their occurrence. A number of authors have shown that abrupt events occur under fully interglacial conditions during MIS 11c (i.e. Barker et al., 2015; Kandiano et al., 2017). The events described here are more similar to those that occurred during the transition from MIS 5e to 5d that are discussed in the introduction. The occurrence of such events, at interglacial/glacial transitions, are relatively well-known, particularly from the North Atlantic, which slightly detracts from the originality of this study. ***The paper needs to consider the rationale and significance of this work in much greater detail, in terms of how it is discussed in the abstract, introduction and conclusion.***

> We agree, there is a growing literature documenting the nature of abrupt events during past interglacial conditions. As clarified above the primary aim of our manuscript is not to prove that high-mag events are possible during low ice conditions but instead to understand how high-magnitude

climate events occur (mechanistically) regardless of the background climate state. Here we advance our understanding of the mechanisms that couple the surface climate with the deep ocean during such events. This is/was also clearly stated in lines 86-92. We will edit the _abstract, introduction, and conclusions_ to reflect our position more clearly.

With regards to the boundary conditions preceding the described events at 397ka and 390ka, we respectfully disagree with the statement above, that both events described occurred "_long after fully interglacial conditions had ceased_".

- At 397ka, radiative forcing received in June at 65N was 466 $W.m^2$ and $CO_2$ concentrations were 259-265 ppmv. Further, the contribution of GIS to global sea levels at 397ka is estimated to have been between 1 and 5 m higher than today (Robinson et al., 2017). MIS 11c at ~397 ka was therefore most similar to preindustrial MIS 1 in terms of eccentricity and $CO_2$ levels at the time (Ganopolski et al., 2016; Yin and Berger, 2015).
- 390ka coincides with a peak in summer insolation at 65°N (Ganopolski et al., 2016) and greenhouse gases were still at or close to interglacial values (CO2: 259.5 ppmv and CH4: 568 ppbv). As highlighted by the reviewer global sea levels are estimated to have been up to 30m lower than today, placing 390ka during an intermediate cryosphere climate. When describing the event at 390ka we do not claim that this even occurs during interglacial boundary conditions. We acknowledge that the sea level lowering that occurred during the precession minima at 395ka would have led to significant ice build-up leading to positive albedo and atmospheric circulation feedbacks.
- Further, we would argue that it is too simplistic to compare the climate backdrop of 397ka to other glacial inceptions based on varying background climate states. The transition from MIS 5e to 5d, for example, was significantly colder than the transition from MIS11c to 11b due to low radiative forcing, see also (Ganopolski et al., 2016).

Secondly, the Rockall trough, as a system, is a hydrographically unique area with cyclonic re-circulation of sediments. Consequently, changes in sediment characteristics within DSDP 610B could reflect variations in the strength of flow within Rockall Trough and not just actual changes in the strength of WTOW. The authors may have considered this but at the moment the manuscript reads as though the complexity of this location is being ignored and overlooked. _The paper needs to show a much greater consideration of the complexities of the oceanographic processes that operate at DSDP 610B and explain why the proxies record the role of WTOW and not more local processes._

We agree with the reviewer's comment and have added information on deep-water flow variability in the oceanographic setting. We have included the following points to better characterise the depositional setting and the potential impact on our records.

- Site 610B is located at the downstream end of the Feni Drift which is a contourite deposit formed by southward flowing of deep water (Ellett and Roberts, 1973; Jones et al., 1970).
- Sediments of the Feni drift are deposited by waning, intermittent bottom currents flowing from North to South along the Feni Ridge because density-driven currents keep bathymetry on their right in the Northern Hemisphere (Johnson et al., 2017). Between episodes of current activity normal pelagic and ice-rafted sedimentation continues unhindered (Robinson and McCave, 1994).
- At the depth of 2417m, 610B lies within the influence of lower WTOW. The flow pathway for WTOW is around the northern and western boundary of the Rockall Trough (Johnson et al., 2017). Observations demonstrate that the southward flow of deep WTOW is intermittent on annual timescales but positive on ≥ decadal timescales (Johnson et al., 2017) which is the resolution that we are targeting in this manuscript.
- To the north and west of 610B the central anticyclonic gyre of the Rockall Trough (Johnson, 2012; New and Smythe-Wright, 2001; Smilenova et al., 2020), recirculates water down to 2000m during winter mixing (Smilenova et al., 2020). Given the distance from the gyre (ca

500km) and the deeper depth of 610B, it is unlikely that it influences the sedimentation and flow over the site.

Thirdly, we are not convinced that the cooling that occurred between 397 and 390 ka should be classified as an "abrupt event". Not only does the event last for some 7,000 years but it is characterised by relatively slow and protracted cooling. This is relatively clearly seen in the SST data presented here where a decline of some 6°C occurs progressively over ca 5 ka.

We argue that the SST record presented here describes a 2-step cooling with a temperature drop of 5.7°C over 1000 years starting at 397.5ka (within dating uncertainties). The onset of this SST cooling, we consider abrupt. Thereafter SSTs are stable for ca 2000 years before a second cooling occurs of 1.8°C over 200 years. Our interpretation for a 2-step rather than a gradual cooling is supported by the ramp function fits shown in figure A2. This method (see also section 2.10) estimates the unknown onset and end of a time interval by weighted least-squares regression by a brute-force search and thereby ensures that onsets and duration of cooling events are chosen objectively. We use a bootstrap simulation of 10,000 resamples to estimate the uncertainty of the results.

The second event that is described is, as the authors acknowledge, is much more typical of an abrupt event (a decline of >6°C in ca 0.5 ka) though this is confidently outside the main interglacial phase and thus does not support their conclusions of high magnitude events during interglacial periods.

As stated above we have edited the abstract, introduction, and conclusions to reflect our findings more clearly.

The change in grain size data for the first event is more dramatic and has much in common with the second event, however, this elicits a very different response in SST values and this is not really acknowledged or addressed.

The phasing of the response between overflows and SST is actually very similar when considering the 2-step nature of the cooling event recorded at 397ka (also described above).

It is also quite important for the authors to discuss the discrepancy in the SST data of late MIS 11c between DSDP 610 and M23414. From 403 ka to 398 ka there is an offset of up to 6₀C between the two sites, significantly greater than the modern temperature gradient between these two localities. Again this isn't discussed but is fundamental to an acceptance of the data and ideas presented here. *The difference between the two events need to be discussed and explored in much greater detail, whilst the validity of the SST estimates for DSDP 610 need to be discussed in more detail, particularly with reference to the record from M23414.*

We thank the reviewer for pointing out this discrepancy and we have investigated this issue.

1.) The SST data from M23414 that we show in Figure 6 was first published in Kandiano et al. (2007). When investigating the methods used to derive SSTs we noted that the data published consists of average values combining three different methods to infer summer SSTs using: Transfer Function Technique (TFT; (Imbrie and Kipp, 1971), Modern Analogue Technique (MAT; (Prell, 1985)) and Revised Analogue Method (RAM; (Waelbroeck et al., 1998)). For all methods SSTs were calculated using the North Atlantic subset of the core-top database as compiled by (Pflaumann et al., 2003). This includes 947 core tops and assumes that the assemblages represent summer SST.

2.) The ForCenS Databased used here, has several advantages over the Pflaumann dataset. (1) It combined four existing key foraminifera compilations (CLIMAP, MARGO, Brown University, & ATL947) and pulls in PANGAEA & NOAA paleoclimatology data. (2) It restricted data synthesis to data generated using the CLIMAP methodology, meaning a minimum count of 300 specimens over 150um (3) ForCenS applied strict procedures to remove duplications and outliers, before going through taxonomic standardization (4) Out of the 6,984 initial records, the database contains 4,205 records from unique sites and informative technical and true replicates.

3.) *The explanation for Offset:* The offset between (Kandiano and Bauch, 2007) and the ForCens generated SST values is mostly due to the fact that ForCens uses the WOA98 ***annual SST*** dataset to generate SST reconstructions, while (Kandiano and Bauch, 2007)

*used summer SST. We argue that the choice of the annual SST dataset is justified particularly during interglacial boundary conditions, because foraminifera bloom twice at subpolar latitudes (Chapman, 2010), once in spring and once in late summer. The assemblages preserved in marine sediments would therefore reflect multiple seasons similarly to the signal preserved in geochemical proxies based on planktonic foraminifera (Leduc et al., 2010). 2010). We corrected the axis information in Figure 6 to Annual SST for our dataset and report Kandiano et al. 2007 as summer SST*

**Figures**: Some general points on uniformity of font sizes, writing (610B – this study) or (610B) or (this study) rather than a combination of the 3

**Figure 1**

Indicating which branch of ISOW is WTOW would be helpful for the reader, particularly as this is the focus of your study. It may also be helpful to include other labels (DSOW, other ISOW branches, NADW etc but not necessary). The grey site label names on a grey background are something I'd advise to change for legibility.

*WTOW is derived from overflows entering the Rockall Trough via the Wyville Thompson Ridge, which saddles the Scottish continental shelf and the Faroe Bank. We included a label on the map to clarify. Grey site labels and names are now in white.*

[Figure]

*Figure 1 Revised Figure 1 with recommendations from Reviewer 1*

**Figure 2**

- In some figures you have labelled the data for this study and in figure 2 you have not. I'm assuming those not labelled relate to DSDP 610 in this study?
  *We have standardized the reference to new data to This study – 610B*
- You've plotted N.incompta + N.deutertrei % together from Kandiano et al 2007 but you've listed this as sub-polar (following Kucera et al 2007), but Deutertrei is a sub-tropical species. You also don't talk about this in the text so wondered why it is plotted?
  *The reason they are plotted together is that they are reported in this way in Kandiano et al. 2007. They are reported in Figure 2 for consistency and information for the reader.*
- The two shades of each colour may be difficult for colour Colour blind people so symbols may help with this.

We added symbols to the Kandiano et al. 2007 data
- The graph feels busy – it might benefit from extending horizontally
  We have extended it horizontally

[Figure]

*Figure 2 Revised Figure 2 as recommended by Reviewer 1*

**Figure 5**
- IRD for 983 is in grey in the axis and appears black in the graph
  We have corrected the colour in the figure caption from grey to black
- The range of font sizes looks untidy
  All font sizes have been unified to size are standardized

[Figure]

**Figure 6**

- IRD for 983 is in grey in the axis and appears black in the graph
  We have corrected the colour in the figure caption from grey to black
- The NPS % from barker et al 2015 was updated in the barker et al 2019 paper. There aren't any major changes in %, though some minor peaks are smaller in the data you have used (e.g., ~ 394ka)
  We have replaced the Barker et al. 2015 with the Barker et al. 2019 dataset.

[Figure]

**Age Model**

Is the age model for ODP 983 tied to DSDP 610B in any particular way or just placed on a timescale with 2 different models (+ associated uncertainties)?

No, the two cores are not tied or tuned to each other and for IODP 983 we are using the AICC2012 as published in (Barker et al., 2015) and in (Barker et al., 2019).

I ask as the LR04 age model for ODP 983 places the increase in NPS % at ~ 395 ka – 389 ka and a second 387-385 ka which is much closer to the authors claims for these events (in both duration and timing). The authors also seem to have tied their core to LR04 so I wonder why (it seems) they have not tied ODP 983 to this.

In Barker et al. 2015 three Age models were proposed for 983 (1) The EDC3 Age model (2) the alternative ice core age model (AICC2012; (Bazin et al., 2013),(Veres et al., 2013)) and (3) an absolute age model (GICC05/NALPS/China) based on (Barker and Diz, 2014; Barker et al., 2011; Cheng et al., 2009). Only the later more dated age model places the increases in NP% at 395 and 385, while the EDC3 and AICC2013 place the first excursion at 390Ka as shown in figure 6.

**Text:**

Some general points line by line through the text. The piece in general would benefit from some sub-headings to organise the flow as presently sections seem to overlap considerably.

- Line 74-75 – would cite Barker et al 2015 "icebergs not the trigger for NA cooling events" (they reference the paper later but they do not cite it here)
  We have added the reference here

- Line 80-84 – quantifying importance of NADW to AMOC and quoting overall contribution from WTOW would be good
  We have added the contribution of WTOW to the total overflow to the manuscript

- Line 90 – Global average temperature difference between MIS 11 and MIS 1 would be good
  According to (Robinson et al., 2017) Summer temperature anomalies at the height of MIS 11 (411ka) were 2.8 ±0.7°C relative to the present. By 403ka, anomalies were at 0 ±1°C relative to the present. By 397ka Summer temperature anomalies had dropped to -2°C.

- Line 109 – add space between reference and 'today'.
  The space was added

- Line 85 – 115. This seems muddled. It seems to be a descriptive piece setting out the conditions during MIS 11 but starts and ends with talk about orbital similarities as a justification for looking at MIS 11. I would have set out the orbital similarities prior to then describing MIS 11.
  This section was restructured according to the reviewer's suggestion.

- Line 347 – 350 – the authors state that the offset is 9 samples (4.5cm), 320 years. Firstly, from the graph, it doesn't look like all 9 of these samples have been run so this is confusing wording
  We assure the reviewer that all 9 samples were analysed and are shown

- Line 404 – random 'o' in the sentence
  This was deleted

- A good paper to cite on surface waters in the Nordic seas being unusually cold

- and fresh in MIS 11 https://www.frontiersin.org/articles/10.3389/fmars.2018.00251/full#h7 which is absent in the bibliography.
  This reference was added

- The reference list needs to be checked there are a number of typos throughout and some repetition (i.e., McManus et al., 1999 is included twice).
  We have checked the reference list throughout.

**END Reviewer 1**

**References used:**

Barker, S., Chen, J., Gong, X., Jonkers, L., Knorr, G., and Thornalley, D.: Icebergs not the trigger for North Atlantic cold events, Nature, 520, 333, 2015.

Barker, S. and Diz, P.: Timing of the descent into the last Ice Age determined by the bipolar seesaw, Paleoceanography, 29, 489-507, 2014.

Barker, S., Knorr, G., Conn, S., Lordsmith, S., Newman, D., and Thornalley, D.: Early interglacial legacy of deglacial climate instability, Paleoceanography and Paleoclimatology, 34, 1455-1475, 2019.

Barker, S., Knorr, G., Edwards, R. L., Parrenin, F., Putnam, A. E., Skinner, L. C., Wolff, E., and Ziegler, M.: 800,000 years of abrupt climate variability, science, 334, 347-351, 2011.

Bazin, L., Landais, A., Lemieux-Dudon, B., Kele, H. T. M., Veres, D., Parrenin, F., Martinerie, P., Ritz, C., Capron, E., and Lipenkov, V.: An optimized multi-proxy, multi-site Antarctic ice and gas orbital chronology (AICC2012): 120-800 ka, Climate of the Past, 9, 1715-1731, 2013.

Chapman, M. R.: Seasonal production patterns of planktonic foraminifera in the NE Atlantic Ocean: Implications for paleotemperature and hydrographic reconstructions, Paleoceanography, 25, PA1101, 2010.

Cheng, H., Edwards, R. L., Broecker, W. S., Denton, G. H., Kong, X., Wang, Y., Zhang, R., and Wang, X.: Ice age terminations, science, 326, 248-252, 2009.

Dokken, T. M., Nisancioglu, K. H., Li, C., Battisti, D. S., and Kissel, C.: Dansgaard-Oeschger cycles: Interactions between ocean and sea ice intrinsic to the Nordic seas, Paleoceanography, 28, 491-502, 2013.

Ellett, D. and Roberts, D.: The overflow of Norwegian Sea deep water across the Wyville-Thomson Ridge, 1973, 819-835.

Ganopolski, A., Winkelmann, R., and Schellnhuber, H. J.: Critical insolation–CO 2 relation for diagnosing past and future glacial inception, Nature, 529, 200-203, 2016.

Imbrie, J. and Kipp, N. G.: A new micropaleontological method for quantitative paleoclimatology: application to a late Pleistocene Caribbean core. In: The Late Cenozoic Glacial Ages, Turekian, K. K. (Ed.), Yale Univ. Press, New Haven, 1971.

Johnson, C.: Tracing Wyville Thomson Ridge overflow water in the Rockall trough, 2012. Aberdeen University, 2012.

Johnson, C., Sherwin, T., Cunningham, S., Dumont, E., Houpert, L., and Holliday, N. P.: Transports and pathways of overflow water in the Rockall Trough, Deep Sea Research Part I: Oceanographic Research Papers, 122, 48-59, 2017.

Jones, E., Ewing, M., Ewing, J., and Eittreim, S.: Influences of Norwegian Sea overflow water on sedimentation in the northern North Atlantic and Labrador Sea, Journal of Geophysical Research, 75, 1655-1680, 1970.

Kandiano, E. S. and Bauch, H. A.: Phase relationship and surface water mass change in the Northeast Atlantic during Marine Isotope Stage 11 (MIS 11), Quaternary Research, 68, 445-455, 2007.

Leduc, G., Schneider, R., Kim, J.-H., and Lohmann, G.: Holocene and Eemian sea surface temperature trends as revealed by alkenone and Mg/Ca paleothermometry, Quaternary Science Reviews, 29, 989-1004, 2010.

New, A. L. and Smythe-Wright, D.: Aspects of the circulation in the Rockall Trough, Continental Shelf Research, 21, 777-810, 2001.

Pflaumann, U., Sarnthein, M., Chapman, M., d'Abreu, L., Funnell, B., Huels, M., Kiefer, T., Maslin, M., Schulz, H., Swallow, J., Kreveld, S. v., Vautravers, M., Vogelsang, E., and Weinelt, M.: Glacial North Atlantic: Sea-surface conditions reconstructed by GLAMAP 2000, Paleoceanography, 18, 1065, 2003.

Prell, W. L.: Stability of low-latitude sea-surface temperatures: an evaluation of the CLIMAP reconstruction with emphasis on the positive SST anomalies. Final report, Brown Univ., Providence, RI (USA). Dept. of Geological Sciences, 1985.

Robinson, A., Alvarez-Solas, J., Calov, R., Ganopolski, A., and Montoya, M.: MIS-11 duration key to disappearance of the Greenland ice sheet, Nature communications, 8, 16008, 2017.

Robinson, S. G. and McCave, I. N.: Orbital forcing of bottom-current enhanced sedimentation on Feni Drift, NE Atlantic, during the mid-Pleistocene, Paleoceanography, 9, 943-972, 1994.

Smilenova, A., Gula, J., Le Corre, M., Houpert, L., and Reecht, Y.: A persistent deep anticyclonic vortex in the Rockall Trough sustained by anticyclonic vortices shed from the slope current and wintertime convection, Journal of Geophysical Research: Oceans, 125, e2019JC015905, 2020.

Thornalley, D. J. R., Elderfield, H., and McCave, I. N.: Holocene oscillations in temperature and salinity of the surface subpolar North Atlantic, Nature, 457, 711-714, 2009.

Veres, D., Bazin, L., Landais, A., Toyé Mahamadou Kele, H., Lemieux-Dudon, B., Parrenin, F., Martinerie, P., Blayo, E., Blunier, T., and Capron, E.: The Antarctic ice core chronology (AICC2012): an optimized multi-parameter and multi-site dating approach for the last 120 thousand years, Climate of the Past, 9, 1733-1748, 2013.

Waelbroeck, C., Labeyrie, L., Duplessy, J. C., Guiot, J., Labracherie, M., Leclaire, H., and Duprat, J.: Improving past sea surface temperature estimates based on planktonic fossil faunas, Paleoceanography, 13, 272-283, 1998.

Yin, Q. and Berger, A.: Interglacial analogues of the Holocene and its natural near future, Quaternary Science Reviews, 120, 28-46, 2015.

---

## Author Comment (AC3)

**Point by point response to Reviewer 2**

We would like to thank Reviewer 2 for helpful comments on our manuscript. Here we have addressed each of the comments and questions in the following format: Each question or comment is re-stated as in the original review of the manuscript in black 'Calibri font'. Our response to each comment/question is indented and written in blue 'Calibri font'.

**General comments:**

The authors are investigating the coupling between surface and deep-water, which is very important. However, I wonder why there is not more comparison with surface condition in the Nordic Seas, as the deep-water at site 610 will be mostly influenced by convection in the Nordic Seas.

- Comparisons with surface conditions in the Nordic Seas over the time interval covered in this manuscript (403-388ka) is difficult because the temporal resolution of available MIS 11 records (e.g. M23352, M23063, and M99-2277/ PS1243) investigated in (Doherty et al., 2021; Doherty and Thibodeau, 2018; Helmke and Bauch, 2003; Helmke et al., 2003; Kandiano et al., 2012; Kandiano et al., 2016) is lower (e.g., 1-3ka per sample).
- However, we will include a short review of the hydrographic conditions in the Nordic Seas in the introduction that will illustrate the boundary conditions present at 403ka.

Moreover, there is a growing body of evidence of the uniqueness of the Nordic Seas during MIS 11 and its key role on North Atlantic circulation. It would be interesting to see how these data compared and a discussion about the mechanism involved. Authors may try to focus on finding mechanistic explanations of the linkages between surface and deep water and go a bit deeper into how this might be relevant to actual climate. What we need to advance the field is a better understanding of the mechanisms behind these critical climatic feedbacks, as we already know very well that interglacials are not *stable*.

We will add a description of the Nordic seas in the introduction and will refer to the potential importance of this unique background state in the discussion (detailed in the above response). However, we also like to affirm here that the main objective of our manuscript is to investigate the phasing between the Atlantic Inflow and deep eastern overflow during climate events following 403ka when MIS11 was on the doorstep to the glacial inception and thus more prone to rapid climate transitions. In our discussion, we are evaluating mechanistic linkages that can explain our observations including the impact of Inflow on the overflows. However, the resolution of records from the Nordic Seas of 1-3ka per sample prevents us from including a meaningful inclusion of the Nordic Seas datasets in our discussion.

I do not understand the choice of timescale investigated here. I would have been interested to see how these data compare from the peak interglacial and the evolution toward the glacial inception rather than focussing only on the 390-399 ka period. This gives a snapshot without much comparable and not much room for more meaningful interpretation of climate evaluation throughout this important climatic period.

Our record begins at 403ka which is the warmest part of MIS 11 when both insolation at 65N was at its peak and the GIS at its minimum (Robinson et al., 2017).

In term of structure, I found the paper rather difficult to follow. I also noted some results and even interpretation presented in the method section, especially in the grain size and chronology section. It makes it more difficult to clearly understand what the original data from this paper are and what is based on literature.

To improve the flow of the paper we place the chronology including the interpretation of the age model prior to the results section so that results can be discussed with reference to time rather than depth. In acknowledgement of the comment by the reviewer, we propose to place the chronology as its own section (3.0) rather than a subsection of the methods so that it is clearly separate.

The results section starts with a general statement discussing the results, I suggest trying to refrain interpretating the results within this section and concentrating the discussion and interpretation in the discussion section. As an example, I would suggest presenting the results in in term of the actual proxy (e.g., grain size variation, foraminifera assemblage) and explain how there are used to reconstruct WTOW and temperature in the discussion section only. Therefore, I recommend a major rework on the structure of the method, results, and discussion sections.

We acknowledge the reviewer's point of view, however, we would prefer to keep the technical description of methods used (e.g., endmember analysis of grain-size distributions and how it is related to deep-water flow) out of the discussion of climate processes and mechanisms. In this way, the discussion is more focused.

**Specific comments:**

The last sentence of the abstract is misleading in my opinion, first because this paper does not provide evidence that the changes, they observed are of similar magnitude than their glacial counterparts.

This has been edited as part of the response to reviewer 1.

Secondly, while this paper might add some evidence, the concept of stable interglacial climate, was already challenged and altered in the past (Bauch, Kandiano, Dickson, etc).

We do not claim that we discover interglacial instability but instead that we contribute to its mechanistic understanding. This will also be clarified in the abstract, introduction and conclusion of the revised manuscript.

Finally, while within an interglacial period, most of the data are not within the peak interglacial, and therefore already within a (long) transition phase as depicted by the ice core data, IRD, etc, so it is not very surprising to see this kind of variability.

See comment above

Introduction: Most of the discussion on deep water is based on the WTOW, please introduce its significance to AMOC, climate, its geometry, etc.

Additional information on WTOW has been added

I would suggest focusing the introduction on the interesting climate feedback that are investigated within the present study and how the uniqueness of MIS 11 rather than the relatively outdated and mostly settled argument on stable interglacials.

We have refocused the abstract and the introduction as recommended by reviewers 1 & 2

L83: Caesar et al., is a brief communication and not a research article per se, I suggest referring to the original research publications e.g., Rahmstorf et al., 2015, etc.

The reference has been changed.

L230 Whole paragraph: It is unclear to me if this should be in the method or results

This paragraph details how we chose the endmembers to reconstruct flow speeds. It is an explanation of why and how we chose end members. For this reason, this paragraph is in the methods section.

L275 Chronology section: I feel most of this should be in the discussion, as the method to acquire the data used here (δ18O, XRF) were already presented. There is a great deal of interpretation to build the chronology.

We place the chronology section prior to the results so that the results can be presented in function of time rather than depth. This helps to focus and streamline the discussion with the results. In acknowledgement to the reviewer, we place the chronology in its own section in the revised manuscript

L330 whole paragraph; not necessary in the results section

We removed the first sentence of this paragraph and focussed the narrative on the results.

L335: One suggestion is to build the results section similarly to the method section, so the reader can easily spot the original data provided by this study. Beware of interpretation the results within this section, the data should be presented here (XRF, grain size, δ18O), but the link to what they are used for (SST, WTOW, etc) should normally go into the discussion.

We prefer to retain the current structure of the results section, which we believe improves the flow of information and interpretation. In this way, the technical interpretation of methods used can be kept out of the discussion of climate processes and mechanisms. We feel that this improves the flow of the discussion and makes it more accessible to a more diverse audience.

L385: I would suggest adding sub-sections to the discussion to try to better structure the arguments that are built here. I find the discussion very hard to follow as of now and I often find myself wondering what exactly the authors want to communicate. I suggest trying to avoid excessive description of published work and instead really focus on new findings.

The discussion of previously published work was a requirement by previous reviewers of this manuscript, and we feel that they provide suitable context to the discussion.

L400: The relationship between fresh and cold water and IRD is not as definitive in the Nordic Seas compared to other regions of the North Atlantic (Doherty et Thibodeau, 2018). See work from Kandiano for salinity reconstructions using alkenones.

As mentioned above we will add a description of the Nordic seas in the introduction and will refer to this unique background state in the discussion to clarify the environmental and climate links.

L485: what would trigger the release of freshwater in a low insolation, cooling climate?

As detailed in response to reviewer 1, at 390ka June insolation at 65N was high and so were global greenhouse gases and we would argue that climate was warming prior to the event. This warming may have triggered the melting of newly built ice over Greenland and potentially even over the LIS.

L485: Is this weakening of the AMOC seen elsewhere? Could it rather be a change in the geometry of the AMOC that leads to this change in your proxy? How does it compare to general trend of AMOC during MIS 11 (e.g., Vazquez Riveiros et al., 2013)

As mentioned in the discussion there is a large body of evidence that the event at 390 was global in character. As for the deep ocean, the event is also seen in benthic 13C records across the North Atlantic (Galaasen et al., 2020; Hodell et al., 2008; Oppo et al., 1998) that we have cited. The event is less evident in records from the deep South Atlantic (e.g., MD07-3077, 3400m water depth; (Riveiros et al., 2013). In Riveiros et al., 2013, the main focus is on the mechanisms and processes that led to Termination V. For the North Atlantic they are also relying on the Oppo 1998 dataset from ODP 980, which as stated above also shows a weakening of the overflows (and their isotopic signature) at 980.

L523: Robinson et al., (2017) model of GIS dynamic also supports this timeframe I believe.

We have added this reference, as suggested

L600-605: maybe it would be good to plot the precession signal somewhere so the reader could appreciate the potential link between these events.

Precession has been added in Figures 5 and 6

L605-610: I am not sure I follow what the authors want to say by: *irrespective of magnitude or boundary conditions* and on what data/evidence this is based.

Here we refer to our observations that the phasing between the surface and the deep is the same regardless of low ice (e.g., 397ka) or intermediate ice (e.g., 390ka) boundary conditions. We will clarify this sentence as part of the revised manuscript.

L620: Rate, volume…what about the location of the freshwater input?

From our dataset, we are unfortunately unable to evaluate the source of fresh water for the event. The location of freshwater could play an important role but it is beyond our study to include it in our discussion.

Figure 2 and 6: the grey bands are not described in the caption (and I believe they are not highlighting the same as in figure 5…)

Corrected as part of revisions made for reviewer 1.

Figure 6: Panel d, there is two datasets listed, but three lines. I suspect this is what is referred to in the last sentence of the caption, but I then I wonder why only one dataset as running average?

We chose to plot a running average for Hodell et al. 2008 in order to compare it to the lower resolution record of Oppo et al. 1998. This is now clarified in the figure caption.

**END of Reviewer 2**

**References used:**

Doherty, J. M., Ling, Y. F., Not, C., Erler, D., Bauch, H. A., Paytan, A., and Thibodeau, B.: Freshening, stratification and deep-water formation in the Nordic Seas during marine isotope stage 11, Quaternary Science Reviews, 272, 107231, 2021.

Doherty, J. M. and Thibodeau, B.: Cold water in a warm world: Investigating the origin of the Nordic seas' unique surface properties during MIS 11, Frontiers in Marine Science, 5, 251, 2018.

Galaasen, E. V., Ninnemann, U. S., Kessler, A., Irvalı, N., Rosenthal, Y., Tjiputra, J., Bouttes, N., Roche, D. M., Kleiven, H. K. F., and Hodell, D. A.: Interglacial instability of North Atlantic deep water ventilation, Science, 367, 1485-1489, 2020.

Helmke, J. P. and Bauch, H. A.: Comparison of glacial and interglacial conditions between the polar and subpolar North Atlantic region over the last five climatic cycles, Paleoceanography, 18, 1036, 2003.

Helmke, J. P., Bauch, H. A., and Erlenkeuser, H.: Development of glacial and interglacial conditions in the Nordic seas between 1.5 and 0.35 Ma, Quaternary Science Reviews, 22, 1717-1728, 2003.

Hodell, D. A., Channell, J. E., Curtis, J. H., Romero, O. E., and Röhl, U.: Onset of "Hudson Strait" Heinrich events in the eastern North Atlantic at the end of the middle Pleistocene transition (∼ 640 ka)?, Paleoceanography, 23, 2008.

Kandiano, E. S., Bauch, H. A., Fahl, K., Helmke, J. P., Röhl, U., Pérez-Folgado, M., and Cacho, I.: The meridional temperature gradient in the eastern North Atlantic during MIS 11 and its link to the ocean–atmosphere system, Palaeogeography, Palaeoclimatology, Palaeoecology, 333, 24-39, 2012.

Kandiano, E. S., van der Meer, M. T., Bauch, H. A., Helmke, J., Damsté, J. S. S., and Schouten, S.: A cold and fresh ocean surface in the Nordic Seas during MIS 11: Significance for the future ocean, Geophysical Research Letters, 43, 10,929-910,937, 2016.

Oppo, D. W., McManus, J. F., and Cullen, J. L.: Abrupt climate events 500,000 to 340,000 years ago: evidence from subpolar North Atlantic sediments, Science, 279, 1335-1338, 1998.

Riveiros, N. V., Waelbroeck, C., Skinner, L., Duplessy, J.-C., McManus, J. F., Kandiano, E. S., and Bauch, H. A.: The "MIS 11 paradox" and ocean circulation: Role of millennial scale events, Earth and Planetary Science Letters, 371, 258-268, 2013.

Robinson, A., Alvarez-Solas, J., Calov, R., Ganopolski, A., and Montoya, M.: MIS-11 duration key to disappearance of the Greenland ice sheet, Nature communications, 8, 16008, 2017.

---

## Author Comment (AC4)

**Point by point response to Reviewer 3**

We would like to thank Reviewer 3 for helpful comments on our manuscript. Here we have addressed each of the comments and questions in the following format: Each question or comment is re-stated as in the original review of the manuscript in black 'Calibri font'. Our response to each comment/question is indented and written in blue 'Calibri font'.

**Specific comments**

The focus of the paper on the concept of interglacial climate stability is somewhat disturbing because the period in which the authors show circulation changes and the presence of IRD does not correspond to the interglacial period of MIS 11 but rather to the glacial inception. The episode starting at 397 ka is associated with the very end of the MIS 11 interglacial period or even marks the beginning of MIS 11b. The paper should be reworked to aim to study the reorganization of the circulation in the high latitudes of the North Atlantic during the glacial inception and not to test the stability of warm climates since the detected circulation changes are not occurring during the course of the interglacial period (even if we remain in an interglacial isotopic stage).

> We edited the abstract, introduction, and conclusions to reflect our findings more clearly – see also the responses to reviewers 1 and 2.

Nevertheless, I agree with the authors that even if the ice volume increases, it is still low during the first NDW reduction interval. If we follow the theory put forward by McManus in 1999, the threshold isotopic value of 3.5 per mil is not exceeded during this interval (it will only be exceeded during the 390 ka episode) which indeed suggests that the ice volume would still be too small to drive millennial events related to ocean-ice feedback mechanisms, leading to changes in the strength of the MOC and changes in interhemispheric heat transport. High latitude circulation changes are apparently happening despite the still low ice volume, but the manuscript does not highlight that the episode between 397 and 392 ka corresponds to a typical millennial-scale event, i.e., an event widely recorded in the North Atlantic SST (from high latitudes to subtropics) and characterized by bipolar see-saw.

> We acknowledge this point made by the reviewer and will include this point in the revised version of the manuscript including relevant references.

The authors may also want to discuss the possibility that the circulation conditions during these 5000 years (that is quite long duration for a millennial event) may reflect orbital variability or processes associated with the glacial inception during the obliquity minimum.

> This is an interesting point, especially since the first reduction of the AMOC at 397ka occurs at a summer insolation minimum (both forced by precession and obliquity), while the second event at 390ka occurs when precession is in the opposite phase, but obliquity is still low.
>
> Related to this topic, Yin et al. 2021 most recently hypothesize that abrupt weakening of the AMOC at the end of interglacial periods (but before glacial boundary conditions are established) is triggered by a combination of a high precession with June solstice occurring at aphelion and, at the same time, a relatively low obliquity (inducing low total summer irradiation). They, propose a sea-ice feedback mechanism to explain the sharp decline in overturning terminating interglacial warmth. Similarly, Zhang et al. 2021 propose a direct astronomical influence on abrupt AMOC variability that is most pronounced during intermediate ice boundary conditions. They propose that either an enhanced boreal seasonality due to a decrease in precession or an increased latitudinal insolation gradient associated with a lowered obliquity can generate a glacial climate background state under which the AMOC oscillates spontaneously. These observations are supportive of our proxy observations at 397ka when precession is high and obliquity low (entering minima), e.g., Yin et al. 2021. The event at 390ka occurs when precession is low, and obliquity is low (exiting minima), which is consistent with Zhang et al 2021 who propose that AMOC variability can be triggered by either precession or obliquity during intermediate cryosphere climate background. The combination of low obliquity and low precession would have led to a strong LTG but also to intense summer heating of

the emergent GIS, which would have led to a warm Nordic Seas and set the stage for the meltwater event at 390ka.

We will include the possibility of orbital forcing for the events in the revised version of our manuscript and thereby contribute to the emerging discussion surrounding this thematic.

The authors cited the paper of Oliveira et al. (2016) mentioning that SST does not record millennial cooling during the 397-397 ka interval but it would have been interesting to note that centennial events are detected in southwestern Europe by the pollen record of the same site (IODP Site U1385, same paper). 3 rapid climatic events showing forest reduction occured at 396 and 393.5 ka without associated SST changes on the Iberian margin and one at 390 associated SST change. It was noted that the first 2 episodes are of lower amplitudes than the 390 ka event, during which the ice volume is larger and for which the other ODP 980 and 983 sites record iceberg discharge episodes. The correspondence is striking enough to be cited, potentially suggesting that the discharge events detected by DSDP site 610B are potentially coupled to atmospheric changes affecting mid-latitude climate.

We agree with the reviewer that the pollen record will make an interesting contribution to the revised version of the manuscript.

**Structure:** Almost all the figures are called in the method section even though they show results or even a comparison with data from the literature. It would seem more appropriate to me to call these figures in the result section. In my opinion, the paragraph between lines 228 and 235 corresponds to results.

We will revise the manuscript so that most of the figures will be called in the results section.

**Chronology:** Did you keep the original age models from core ODP 983 and U1308? It is difficult to justify that it is better to correlate with a record from the same region (whose age model is based on Martinson (1987) benthic isotope stack) instead of LR04 and then compare with the benthic records from those sites, one of which based on EDC 3 age model and the other on LR04 and later, mention that there is a good correlation between the records. This procedure is lacking of consistency.

For ODP 983 we used the age model published in Barker 2015 and 2019 – see also comment for reviewer 1, and for U1308 we used the age model published by Hodell et al. 2008 (based on LR04)

**Technical corrections**

Summer insolation and astronomical parameter curves should be added in Figure 5 and/or 6.

This is now added to Figures 5 and 6

The limit of the isotopic substages should be indicated in at least one figure.

We have added the isotopic substages (as defined in (Railsback et al., 2015) in Figure 6

The correction of isotopic values used to present the ODP 980 & 983 and U1308 sites in Figure 4 should be indicated.

The correction of 0.63 is now added to the figure caption of Figure 4

May be great to add in Figure 6 the curve of NPS % from site DSDP 610 with those of ODP Site 983.

This is now added to Figure 6

328: Beginning the results section with a sentence mentioning the similarity to the Holocene does not seem appropriate.

We have removed this sentence also as part of revisions requested by Reviewer 2

404: "Further our results".

This has been changed

473: Values of CO2 and

This has been revised

**References used:**

Railsback, L. B., Gibbard, P. L., Head, M. J., Voarintsoa, N. R. G., and Toucanne, S.: An optimized scheme of lettered marine isotope substages for the last 1.0 million years, and the climatostratigraphic nature of isotope stages and substages, Quaternary Science Reviews, 111, 94-106, 2015.

---

## Referee Report (RR1)

**Review summary**

**Recommendation: Minor Revision**

Firstly, I would like to welcome the revised manuscript. The reviewers have overall addressed many of the reviewer's comments. The structure is much improved and, in particular, the premise is much better. There are, however, 3 outstanding issues remaining that need to be addressed prior to publication. There are further minor issues I think the author should address.

**Major comments**

   **A) Chronology**

In the initial round of reviews, the chronologies of cores used for data comparison was raised. The authors replied to this in the comments stating which age model they used for ODP 983 (AICC 2012). The reply comments did not sufficiently address this, and the handling editor requested this be included as a discussion point. This does not invalidate the data of the study so there is certainly scope for addressing this; however, it is important to some of the arguments made in the discussion (in particular 5.2).

There are numerous age models in the literature used for ODP 983 (LR04, Lisecki and Raymo 2004; adjusted LR04 Wolf et al., 2011; AICC 2012 / EDC 3 (Barker et al., various); and U1385 (Barker et al., 2021). The authors have chosen AICC 2012, placing the 1$^{st}$ event in ODP 983 between 391 and 387 ka. There is no increase in ODP 983 NPS % during the timing of the authors first event in core 610 in their 983 age model (AICC 2012). This is similar for the EDC3 age model (391 – 386). LR04; however, places the onset earlier event in ODP 983 at ~ 393.5 – 386 and the 2021 U1385-based model for 983 puts an event at 396 – 391. I acknowledge that this is more problematic when it comes to the second event at 983, which is much later in LR04 than all of the models (though, notably much earlier in in the U1385 based model).

There are significant difficulties in comparing ocean cores due to the uncertainties associated with age models during MIS 11c (quoted +/- 4ka, as the authors acknowledge). As a result, the authors are not wrong to choose AICC 2012 as an age model; however, given they are using LR04 ties for their site (610) I'm curious to why they haven't chosen the LR04 age model for ODP 983; particularly, as this fits with (at least the first, and longest event in) their data much better than the chosen model. Again, this is not to say the authors are wrong but the choice of age model could result in different arguments being made and must be mentioned.

This is important because the absence of their 1st event at ODP 983 is an important part of their hypothesis for north Atlantic dynamics at the time. I have included a quick graph on the differences in the choice of age model make to the '1st' (black arrows) and '2nd' (red arrows) event in ODP 983. **The authors should include either / both of the following:**

1. **A justification for the age model chosen over others for comparison data (e.g. EDC3, LR04, U1385) with a brief section addressing that there may be other interpretations to be made with different age models.**
2. **A dual discussion point, proposing theories for each age model outcome (e.g. ocean dynamics in the circumstances of a 1st event / without a 1st event at ODP 983).**

[Figure]

Addressing this in text is essential to the manuscript being published. I want to re-iterate this is not a criticism of the author's choice of age model but stating that a justification for that choice is currently absent from the text. At present, I would argue that the first event in ODP 983 has far more in common with the first event in your data (610) and that the 2 events actually line up well with those in ODP 983 (indeed, the IRD peaks would also line up quite well), but that is difficult to say without younger data to see what occurs after your two events (see **C**).

**B) Study premise**

Overall, the authors have well addressed the initial paper's issue where it seemed to be discussing abrupt climate events during interglacials as a new phenomenon. I thank the authors for clarifying this in their revised manuscript. I also agree with the authors that the 1st of these events is occurring during low-cryosphere climate and that itself is important. That said, I still would not agree that the event is occurring within the main interglacial phase (despite GHG emissions / low cryosphere climate) but is happening as part of the start of the transition into glacial inception. **As such, there are a couple of areas where I think the authors should clarify the wording:**

1. Line 70-74: The statement of "similar to today" may be problematic. This is in the glacial inception/post main interglacial phase. This is at a period of ice sheet growth rather than ice sheet reduction. You do address this later on; however, so this may be something to address structurally rather than changing the wording (see 'minor comments' for more).

2. Line 430-431: You say previously it was only observed in glacial boundary conditions, but these are glacial/interglacial boundary conditions (albeit low ice volume). I am unsure if you are referring to a direction of travel (e.g. previously only seen in glacial – interglacial rather than in this case, interglacial to glacial). If so, this needs clarifying.

**C) 2ⁿᵈ Abrupt event**

At present, the authors do not have data after ~ 389 ka BP. As such, how confident are the authors that that event has 'finished'? The authors are making assertions about leads and lags between SSTs and deep ocean circulation, which are unaffected by this, but the duration of recovery may be. The recovery seems clear and complete in the raw NPS % data but less clear in the MAT and WTO data. Doing low resolution sampling on younger samples in the core might help this and also help clarify whether there is an absence of a 1ˢᵗ event in ODP 983 or whether it is a function of the age model chosen. **The authors should include either:**

1. **More data on younger samples to fully characterise the end of their 2ⁿᵈ event**
2. **Should this not be possible, a brief point explaining that younger data may impact the absolute recover times, particularly in the WTO dataset.**

**Minor comments**

1. I think the abstract could better show off your data as a summary. At present it doesn't mention the site nor the identification of 2 events in the data (the latter of which I think would be quite eye catching for the casual reader scrolling through!)

2. Line 58: Forcing has a strikethrough – is this intentional?

3. The structure of the introduction could be improved. Line 70-74: The statement of "similar to today" may be problematic. This is in the glacial inception/post main interglacial phase. This is at a period of ice sheet growth rather than ice sheet reduction. You go to discuss this later on – I suggest the authors re-arrange this to be prior to your research aim. Presently, as a reader I think about why you have said this and then that doesn't immediately get clearly get answered and what is there is much later on. I would put your research justification for MIS 11 vs MIS 1 prior to the aim of your study. I'm not confident I agree with the 'similar to today' comment and would suggest changing it to "during low cryosphere climate states" but I think it would be more fine for the reader to see you go from (1) high vs low magnitude variability (2) need to study low cryosphere climate states to investigate this (3) justifying MIS 11c glacial inception as an important area of study (**and crucially, the time slice you have chosen**) (4) therefore the aim of your study is to investigate site 94-610, which will flow well into your methods.

4. It is good you have clarified the conditions of Rockall Trough and how your site is not impacted by any issues associated with this. As a suggestion but not necessarily mandatory - I wonder if there is data to show in other interglacials (e.g. Holocene, MIS 5e) that sites of similar proximity / depth have also not experienced these issues that you could reference?

5. Line 424: you reference figure 6 and refer to a number of sites in preceding lines. One of these is 1063, but 1063 is not in your figure 6. I see figure 5 has the C13 data

in a longer timespan but it is focussed on your time slice in figure 6. Did you mean to add 1063 to figure 6 or are you wanting to refer to figure 5 instead?

---

## Author Response (AR2)

We thank Reviewer 2 for final comments on our manuscript. Below we have addressed all outstanding issues.

**A) Chronology:**

As pointed out by the reviewer many age models have been presented for ODP site 983 over the years. We chose the AICC2012 age model for 983 because we agree with the assumption that abrupt warming events are likely synchronous with warming across the wider North Atlantic region (Austin and Hibbert, 2012; Barker et al., 2015; Hodell et al., 2013) and with peaks in atmospheric methane (Govin et al., 2012) recorded in the EPICA Dome C ice core. We state the rationale for this assumption in the Chronology Section. "*Following the age model approach used by Govin et al. (2012), we compared peaks in atmospheric methane recorded in the EPICA Dome C ice core record to our chronology for DSDP 610B ; adopting the same assumption that abrupt warming events are synchronous with warming across the wider North Atlantic region (Hodell et al. 2013; Austin et al. 2012; Barker et al. 2015).*"

We therefore believe that the EDC3 or the AICC2012 chronology presents an improvement for 983 (used in Barker et al. (2019) and Barker et al. (2015) over the original LR04 model based on 80 tie points (one used every ~22 ka, on average) over the 1.8-million-year record (Raymo et al., 2004). We would be hesitant to choose LR04 just because the dataset "fit better" with our results. However, as the reviewer notes, it is important to state the rationale for these assumptions and how the remaining uncertainty impacts the interpretation. In response to these two points we have updated both the chronology and discussion sections to address this.

In the revised manuscript we have clarified in ll 337-343: The 390 ka methane event also corresponds to a coeval enrichment of benthic oxygen isotope values recorded in DSDP 610 and ODP 980 and 983 (EDC3 Age model (Barker et al., 2015)) and to a rapid cooling event (Uk'37-SST) recorded further south off the Iberian continental margin (Oliveira et al., 2016) (International Ocean Drilling Project (IODP) Site U1385). *Based on this agreement we chose to plot data from ODP 983 according to the AICC2012 chronology as in Barker et al. (2015) and Barker et al. (2019) in favour over the LR04 chronology from 2004 (Raymo et al. 2004). For the time interval of interest, the two chronologies (e.g., AICC2012 and LR04) for ODP 983 are well within the ±4 ka uncertainty. However, this dating uncertainty does affect the certainty with which the relative timing of events described in this manuscript can be interpreted. Notably, the age models used here result in an excellent alignment of the benthic $\delta^{18}O$ enrichment event at 390 ka (Figure 3) which is a distinct feature at all sites (DSDP 610, ODP 980 and 983).*

We also modify the wording in the discussion of the 397ka event to highlight the dependence of the interpretation on the age model in section **5.2. Reorganization of Atlantic Waters in the SPG** *(ll. 475-480)*

In the wider palaeoceanographic context, the onset of sea surface cooling observed at our site over the Rockall Trough at ca. 397 ka also occurs at site M23414 200 km west of DSDP site 610 (Fig 6b; Kandiano and Bauch (2007)) but is not evident further west, closer to the SPG (ODP site 983, Gardar Drift 60.48 N, 23.68 W), where Neogloboquadrina pachyderma abundances remain low and stable (0–10%) (see also Fig. 6) until ca. 391 ka (Barker et al., 2015). *While we cannot strictly rule out that the 391ka cooling observed at ODP 983 is actually occurring at 397 ka, most age models published for this site (EDC3, AIC2012; Barker et al. 2015) place this event around 391 ka (except LR04; Raymo et al. 2004, which places it closer to 393 ka) and this timing is supported by the close alignment of the distinct benthic d18O excursion at all North Atlantic sites (Figure 3; cf chronology section) and the methane changes at this time (Figure 4). In the absence of changes at the edge of the SPG (ODP 983),* it is

unlikely that a major displacement of oceanic fronts, as found by Irvalı et al. (2016) and Mokeddem et al. (2014) to mark the demise of the last interglacial, occurred at this time.

**B) Study Premis:**

Line 70-74: The statement of "similar to today" may be problematic. As suggested by the reviewer we have changed this wording to: *"during low cryosphere climate states."*

Line 430-431: You say previously it was only observed in glacial boundary conditions, but these are glacial/interglacial boundary conditions. We have clarified that we refer to *"intermediate or large cryosphere boundary conditions"*

**C) 2nd Abrupt event**
At present, the authors do not have data after ~ 389 ka BP. As such, how confident are the authors that that event has 'finished'? Unfortunately, more data aside from the XRF is not available and not feasible to produce at this stage. In lines 400-401 we added the following sentence to acknowledge the uncertainty linked to the recovery: *Since WTOW values do not stabilize by the end of the timeseries produced for this study the recovery may extend beyond 388.5 ka*

**Minor Comments:**

1. I think the abstract could better show off your data as a summary. At present it doesn't mention the site nor the identification of 2 events in the data (the latter of which I think would be quite eye catching for the casual reader scrolling through!) We have added the site and reference to two climate events

2. Line 58: Forcing has a strikethrough – is this intentional? No this wasn't intentional just an oversight. We have deleted the word in the revised manuscript.

3. The structure of the introduction could be improved. We thank the reviewer for his/her recommendations; however, we feel that the restructuring is a subjective preference of the reviewer rather than a structural necessity that will improve the flow of the manuscript.

4. It is good you have clarified the conditions of Rockall Trough and how your site is not impacted by any issues associated with this. As a suggestion but not necessarily mandatory - I wonder if there is data to show in other interglacials (e.g., Holocene, MIS 5e) that sites of similar proximity / depth have also not experienced these issues that you could reference? We are not aware of any study not already mentioned in the manuscript that would specifically be able to demonstrate that their sedimentary record from the Feni Drift is not impacted by recirculation of southern sourced deep waters.

5. Line 424: you reference figure 6 and refer to a number of sites in preceding lines. One of these is 1063, but 1063 is not in your figure 6. I see figure 5 has the C13 data in a longer timespan but it is focussed on your time slice in figure 6. Did you mean to add 1063 to figure 6 or are you wanting to refer to figure 5 instead? To ensure that all datasets are referred to we included references to both figures here.

**References:**

Austin, W. E. and Hibbert, F. D.: Tracing time in the ocean: a brief review of chronological constraints (60–8 kyr) on North Atlantic marine event-based stratigraphies, Quaternary Science Reviews, 36, 28-37, 2012.

Barker, S., Chen, J., Gong, X., Jonkers, L., Knorr, G., and Thornalley, D.: Icebergs not the trigger for North Atlantic cold events, Nature, 520, 333, 2015.

Barker, S., Knorr, G., Conn, S., Lordsmith, S., Newman, D., and Thornalley, D.: Early interglacial legacy of deglacial climate instability, Paleoceanography and Paleoclimatology, 34, 1455-1475, 2019.

Govin, A., Braconnot, P., Capron, E., Cortijo, E., Duplessy, J. C., Jansen, E., Labeyrie, L., Landais, A., Marti, O., Michel, E., Mosquet, E., Risebrobakken, B., Swingedouw, D., and Waelbroeck, C.: Persistent influence of ice sheet melting on high northern latitude climate during the early Last Interglacial, Clim. Past, 8, 483-507, 2012.

Hodell, D., Crowhurst, S., Skinner, L., Tzedakis, P. C., Margari, V., Channell, J. E., Kamenov, G., Maclachlan, S., and Rothwell, G.: Response of Iberian Margin sediments to orbital and suborbital forcing over the past 420 ka, Paleoceanography, 28, 185-199, 2013.

Irvalı, N., Ninnemann, U. S., Kleiven, H. K. F., Galaasen, E. V., Morley, A., and Rosenthal, Y.: Evidence for regional cooling, frontal advances, and East Greenland Ice Sheet changes during the demise of the last interglacial, Quaternary Science Reviews, 150, 184-199, 2016.

Kandiano, E. S. and Bauch, H. A.: Phase relationship and surface water mass change in the Northeast Atlantic during Marine Isotope Stage 11 (MIS 11), Quaternary Research, 68, 445-455, 2007.

Mokeddem, Z., McManus, J. F., and Oppo, D. W.: Oceanographic dynamics and the end of the last interglacial in the subpolar North Atlantic, Proceedings of the National Academy of Sciences, 111, 11263-11268, 2014.

Oliveira, D., Desprat, S., Rodrigues, T., Naughton, F., Hodell, D., Trigo, R., Rufino, M., Lopes, C., Abrantes, F., and Goni, M. F. S.: The complexity of millennial-scale variability in southwestern Europe during MIS 11, Quaternary Research, 86, 373-387, 2016.

Raymo, M. E., Oppo, D. W., Flower, B. P., Hodell, D. A., McManus, J. F., Venz, K. A., Kleiven, K. F., and McIntyre, K.: Stability of North Atlantic water masses in face of pronounced climate variability during the Pleistocene, Paleoceanography, 19, 2008, 2004.